# Snow or rain? hybrid AI deciphers surface precipitation phase from satellite observations

Chunlei Yang[1,2,6], Haoran Li[3,6], Runzhe Zhu[1,2], Yan Wang[4], Feng Zhang [1,5] ✉, Mingjian Gu[4], Geng-Ming Jiang[2], Renhe Zhang [1,5] & Xu Tang [1,5]

Surface precipitation phase transition is conducive to devastating snowstorms and avalanches yet remains a global challenge due to the scarcity of surface observations. Here, we present the Real-time Precipitation Phase-Intensity Collaborative Retrieval Network (RePPIC-Net), a hybrid AI framework that quantifies surface precipitation phase from satellite observations. By integrating real-time 3D atmospheric physics fields from the AI-driven FuXi model with operational geostationary satellite observations through a hierarchical architecture, our system enables real-time monitoring of surface precipitation phase, as opposed to at least 4-hour latency of current operational systems. Validated against ground stations in China, RePPIC-Net achieves a Critical Success Index for Phase and Detection of 0.1574 (snowfall) and 0.3147 (rainfall) for 0.1-5 mm/h precipitation, outperforming 4-hour latency operational products' respective scores of 0.1001 and 0.3064. The real-time precipitation phase discrimination capability of RePPIC-Net allows the development of a satellite-based surface precipitation phase nowcasting system, meeting the need for 1-3 hour global surface precipitation phase transition warnings. RePPIC-Net provides a replicable blueprint for AI-powered real-time weather monitoring, filling a gap in wintertime weather disaster warnings.

Accurate discrimination of precipitation phase is pivotal for mitigating cascading meteorological disasters, particularly in the context of accelerating climate change. Subtle phase shifts trigger divergent societal risks: transient snow-to-rain transitions destabilize snowpack structures, elevating avalanche hazards in alpine zones[1-4], while rain-to-snow conversions under cooling anomalies amplify blizzard risks, paralyzing critical infrastructure[5-8]. The societal stakes are exemplified by China's 2024 snow disasters, which affected 9.07 million people and claimed 24 lives through transport gridlock and supply chain failures[9]. Climate warming exacerbates these threats by intensifying phase volatility across high-latitude and high-altitude regions—precisely where ground-based monitoring is most deficient due to wind-induced gauge undercatch and sparse station density[10]. While satellite systems like the Global Precipitation Measurement (GPM) mission's Integrated Multi-satellite Retrievals (IMERG) product provide global coverage, their operational latency (>4 h) and coarse spatiotemporal resolution (>0.1°/30 min)[11,12] fail to resolve rapid phase transitions given the time-costing post-processing satellite observations and traditional numerical weather prediction (NWP) models[13]. This system void undermines real-time hazard response, highlighting an urgent need for phase detection frameworks tailored to warming-sensitive terrains.

[1]Key Laboratory of Polar Atmosphere-Ocean-Ice System for Weather and Climate of Ministry of Education, Department of Atmospheric and Oceanic Sciences, Fudan University, Shanghai, China. [2]College of Future Information Technology, Fudan University, Shanghai, China. [3]State Key Laboratory of Severe Weather Meteorological Science and Technology, Chinese Academy of Meteorological Sciences, Beijing, China. [4]National Key Laboratory of Infrared Detection Technologies, Shanghai Institute of Technical Physics, Chinese Academy of Sciences, Shanghai, China. [5]Integrated Research on Disaster Risk International Centre of Excellence, Fudan University, Shanghai, China. [6]These authors contributed equally: Chunlei Yang, Haoran Li. ✉e-mail: fengzhang@fudan.edu.cn

Artificial intelligence offers distinct methodological paradigms for precipitation monitoring and nowcasting[14–17], yet current technical frameworks face dual challenges of real-time multi-source data fusion and insufficient observational representativeness[18–20]. Although deep learning models such as UNet can achieve precipitation retrieval at 5 km resolution by integrating multi-spectral satellite data[21], their purely data-driven approaches particularly result in intensity retrieval errors in complex terrain[10]. While active radars and microwave sensors on polar-orbiting satellites can detect precipitation phase transitions by analyzing vertical hydrometeor profiles through reflectivity variations, current satellite-based phase retrieval still demonstrates significant bias compared to NWP-based phase results[22–26]. Especially, geostationary satellites lack full-disk vertical detection capabilities and thus fail to produce spatial-continuous phase-resolved precipitation products[27,28]. Meanwhile, meteorological forecast models (e.g., FuXi) generate physically consistent 3D atmospheric fields via Transformer architecture[29,30], outperforming traditional numerical models in synoptic-scale circulation forecasting[31–33]. However, the absence of multi-source observations and inadequate parameterization of mesoscale dynamical processes hinder their ability to accurately resolve small-to-medium-scale precipitation events[34–36]. This disconnect between vertical meteorological background fields with high-resolution real-time satellite observations leaves existing operational systems incapable of concurrently addressing critical requirements for enhanced spatiotemporal resolution, improved precipitation phase identification accuracy, and time-sensitive disaster alert capabilities.

To address this, we propose Real-time Precipitation Phase-Intensity Collaborative Retrieval Network (RePPIC-Net)—a hybrid framework integrating 3D meteorological physical fields (generated within seconds) with high-resolution satellite observations, establishing a real-time operational system enabling concurrent phase-resolved precipitation monitoring. Also, an operational nowcasting framework that seamlessly integrates satellite brightness temperature forecasts from a satellite nowcasting model to generate precipitation predictions with explicit rain-snow differentiation—a previously unmet need bridged by overcoming GPM IMERG's reliance on lagged NWP assimilation for phase quantification. Validated by ground stations across China, RePPIC-Net demonstrates superior snow identification (Critical Success Index for Phase and Detection = 0.1574 vs. GPM IMERG's 0.1001) at 0.1–5 mm/h precipitation sensitive range. This work not only provides a real-time monitoring tool for data-sparse regions but also establishes a technological paradigm to advance global early warnings.

## Results

### Evaluations of RePPIC-Net based on ground observation

As shown in Fig. 1a, a nationwide precipitation phase evaluation of RePPIC-Net against the GPM IMERG-Late product, utilizing over 2000 national Automatic Weather Stations (AWS) across China, revealed comparable rain-detection performance between the models with Critical Success Index (CSI) scores of 0.85 and 0.83, respectively. In addition, both models exhibited slight snowfall overestimation (Bias > 1.0) and rainfall underestimation (Bias < 1.0), with RePPIC-Net (snow:1.02; rain:0.99) closer to 1.0 than GPM IMERG (snow:1.36; rain:0.84). However, RePPIC-Net exhibited moderately reduced skill in snow identification (CSI: 0.69 vs. GPM IMERG's 0.71), driven by elevated false-alarm rates at localized stations.

To further assess the monitoring capability for mixed-phase precipitation, a targeted analysis was performed for weak precipitation events within the 0.1–5 mm/h intensity range. The threshold is operationally significant, as snowfall in this category often corresponds to heavy or blizzard conditions, whereas rainfall represents light to moderate events. As illustrated in Fig. 1b, GPM IMERG outperformed RePPIC-Net in snowfall phase identification metrics, whereas RePPIC-Net excelled in snowfall detection accuracy (Critical Success Index for Detection, $CSI_D^n$: 0.23 vs. 0.14). Meanwhile, for rainfall, RePPIC-Net

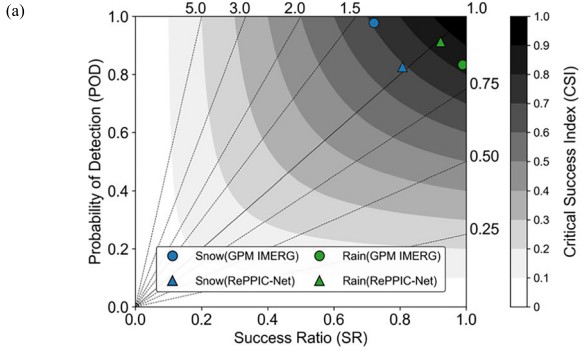

(a)

(b)

| | | GPM IMERG | | RePPIC-Net | |
|---|---|---|---|---|---|
| | | CSI | Uncertainty(±) | CSI | Uncertainty(±) |
| Phase | Rain | 0.8252 | 0.0004 | 0.8477 | 0.0004 |
| $(CSI_P)$ | Snow | 0.7099 | 0.0005 | 0.6896 | 0.0007 |
| Detection | Rain | 0.3713 | 0.0017 | 0.3713 | 0.0018 |
| $(CSI_D^n)$ | Snow | 0.1411 | 0.0034 | 0.2283 | 0.0041 |
| Overall | Rain | 0.3064 | 0.0014 | **0.3147** | 0.0015 |
| (CSI-PD) | Snow | 0.1001 | 0.0024 | **0.1574** | 0.0028 |

**Fig. 1 | Evaluation of precipitation detection and phase for RePPIC-Net and GPM IMERG-Late. a** Overall performance diagram showing the Probability of Detection (POD), Success Ratio (SR), Bias, and Critical Success Index (CSI) for RePPIC-Net and GPM IMERG-Late in discriminating between rainfall and snowfall. The dashed lines denote bias scores with labels on the outward extension of the line. **b** Critical Success Index for phase ($CSI_P$), Critical Success Index for Detection ($CSI_D^n$), Critical Success Index for Precipitation Phase and Detection (CSI-PD) and uncertainty of RePPIC-Net and GPM IMERG within the 0.1–5 mm/h precipitation intensity range.

demonstrated better phase discrimination capabilities, with both models performing similarly in precipitation detection. The consistently low uncertainties across all metrics indicate robust evaluation results, though detection CSI shows marginally higher uncertainty than phase classification. The superior comprehensive performance of RePPIC-Net, integrating spatiotemporal consistency and operational applicability, underscores its enhanced utility for monitoring small-scale snowfall systems. These findings suggest that while phase-specific algorithms in GPM IMERG provide robust discrimination, RePPIC-Net's improved detection sensitivity aligns more closely with real-time disaster mitigation requirements, particularly in regions prone to rapid-onset winter storms.

### The spatiotemporal error distributions

The national AWS observations allowed an examination of spatiotemporal error distribution characteristics of RePPIC-Net (Figs. 2 and 3). In general, both models demonstrated consistent spatial error distribution patterns, characterized by a relatively diminished capacity for rain discrimination over the Tibetan Plateau and a reduced ability to discriminate snow in southern China. Regarding rainfall, RePPIC-Net performed slightly better than GPM in the northwestern and northeastern regions with higher CSI (Fig. 2a, b) and POD (Fig. 2e, f). Nevertheless, this improvement was accompanied by a certain increase in False Alarm Ratio (FAR) (Fig. 2i, j). Regarding snowfall, both models were generally comparable; however, RePPIC-Net showed inferior performance in southern areas (Fig. 2c, d) due to lower POD (Fig. 2g, h), indicating that the model tends to miss snowfall in mountainous and similar terrains. These findings underscore the need for terrain-aware parameterizations and enhanced thermodynamic constraints in solid-precipitation retrievals.

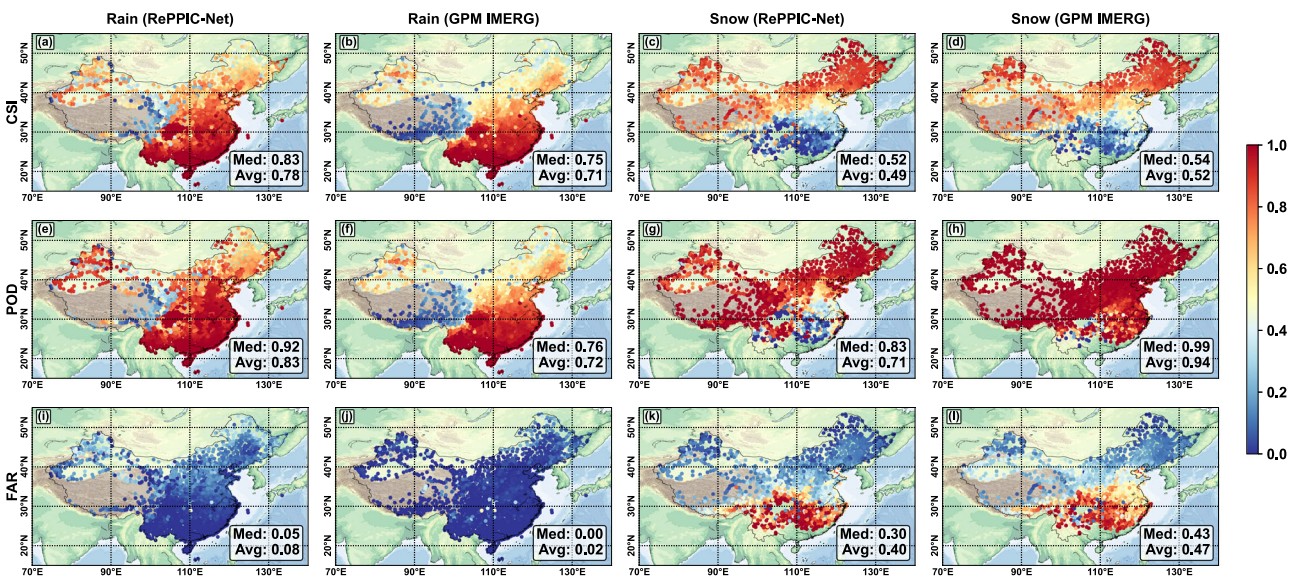

**Fig. 2 | Evaluation of precipitation phase.** Spatial distributions of the (**a**–**d**) Critical Success Index (CSI), (**e**–**h**) Probability of Detection (POD), and (**i**–**l**) False Alarm Ratio (FAR) for rain and snow from RePPIC-Net and GPM IMERG-Late.

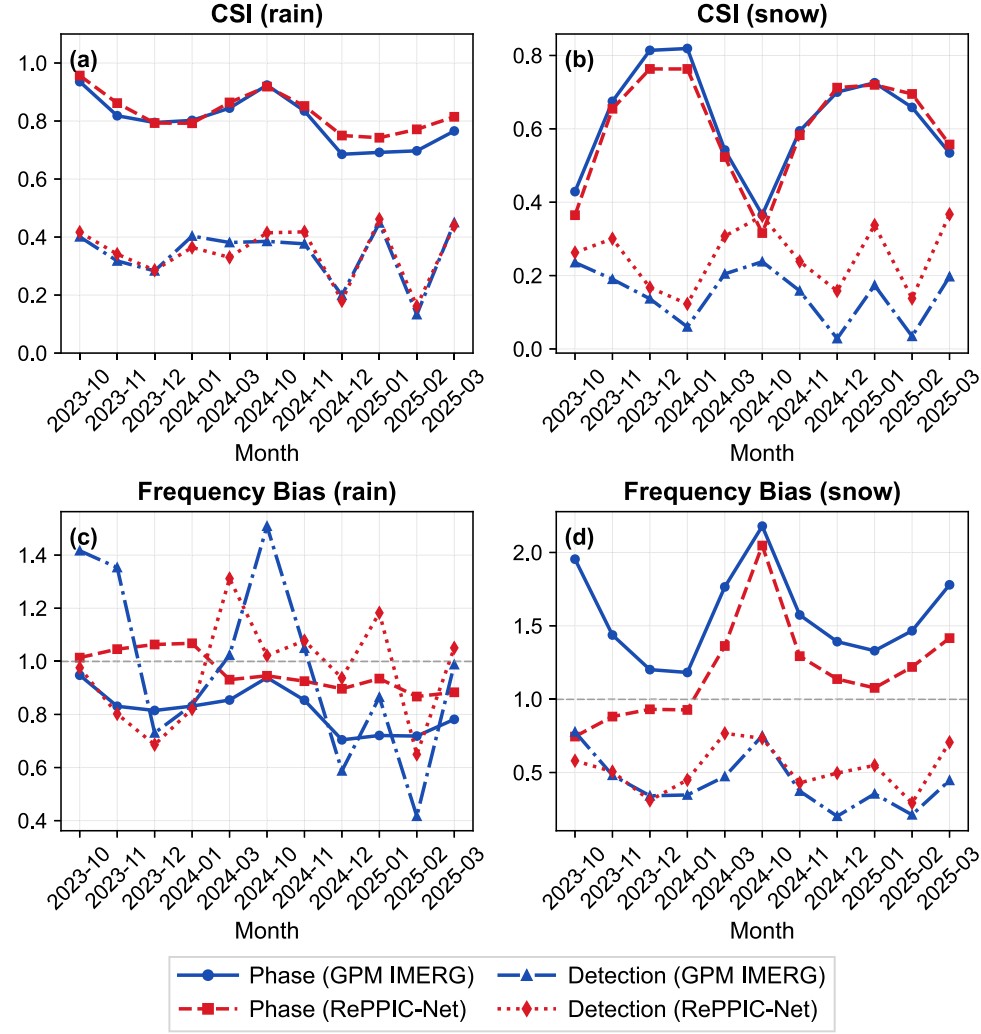

**Fig. 3 | Temporal variations of Critical Success Index (CSI) and Frequency Bias for RePPIC-Net and GPM IMERG-Late.** CSI performance is shown for rain in (**a**) and snow in (**b**) across different months. **c**, **d** show monthly temporal variations of Frequency Bias for the two products (rain and snow, respectively).

Analysis of temporal variations further revealed the stability of model performance (Fig. 3). The study found that the discrimination results for snow phase exhibit significantly greater variation over time than those for rain, following certain patterns: rain shows lower CSI values in December and January, potentially related to the complex phase discrimination of winter rainfall in southern China; whereas snow demonstrates notably lower CSI in October and March, indicating that the model still faces challenges in accurately identifying phase during transition periods at the beginning and end of the snow season. In terms of monthly trends, RePPIC-Net and GPM IMERG show similar variation patterns, but the former consistently tends to classify more precipitation as snow across all months, correspondingly reducing rain classifications than the latter. Regarding overall precipitation detection capability, both models show relatively stable CSI performance for rain and snow across different months, with RePPIC-Net demonstrating superior snow detection ability throughout all months.

## Case studies of mixed-phase and orographic snowfall

Under the influence of a strong cold air mass, a high-impact mixed precipitation event occurred across northeastern China from 20:00 UTC on 5 November 2023 to 20:00 UTC on 7 November 2023. The event featured heavy snowfall transitioning to rain-snow mixtures, with accumulations ranging from heavy snow (10–20 mm/day) to localized blizzard conditions (>30 mm/day) in central and western Inner Mongolia, northern and central Heilongjiang, Jilin, and Liaoning provinces. The China Meteorological Administration (CMA) issued a blue cold wave alert and an orange blizzard alert on 5 November. Joint consultations between the Ministry of Emergency Management and CMA emphasized preparedness for rapid temperature drops, ice accumulation, and complex precipitation phases driven by abundant moisture and dynamic cooling.

Quantitative assessments based on ground station data revealed significant discrepancies among satellite precipitation products (Fig. 4a). The official Fengyun-4B (FY-4B) precipitation product exhibited substantial overestimation with notable spatial distribution errors. The GPM IMERG product also overestimated total precipitation, while RePPIC-Net demonstrated superior accuracy, achieving the lowest magnitude error (RMSE: 1.65 mm/h). Critical to this event was the ability to distinguish rain from snow, particularly in regions where near-freezing temperatures caused rapid phase transitions. Both RePPIC-Net and GPM IMERG incorporate phase discrimination modules, enabling monitoring of snowfall. RePPIC-Net outperformed GPM IMERG in detecting blizzard conditions over northern China, attributed to its higher spatial resolution. For snowfall detection, RePPIC-Net achieved a Critical Success Index for Precipitation Phase and Detection (CSI-PD) of 0.64, surpassing GPM IMERG (CSI-PD: 0.39). For rainfall detection, RePPIC-Net also exhibited a slight advantage over GPM IMERG (CSI-PD: 0.49 vs. 0.43), suggesting its deep learning architecture effectively captures precipitation patterns despite GPM's multi-sensor fusion approach.

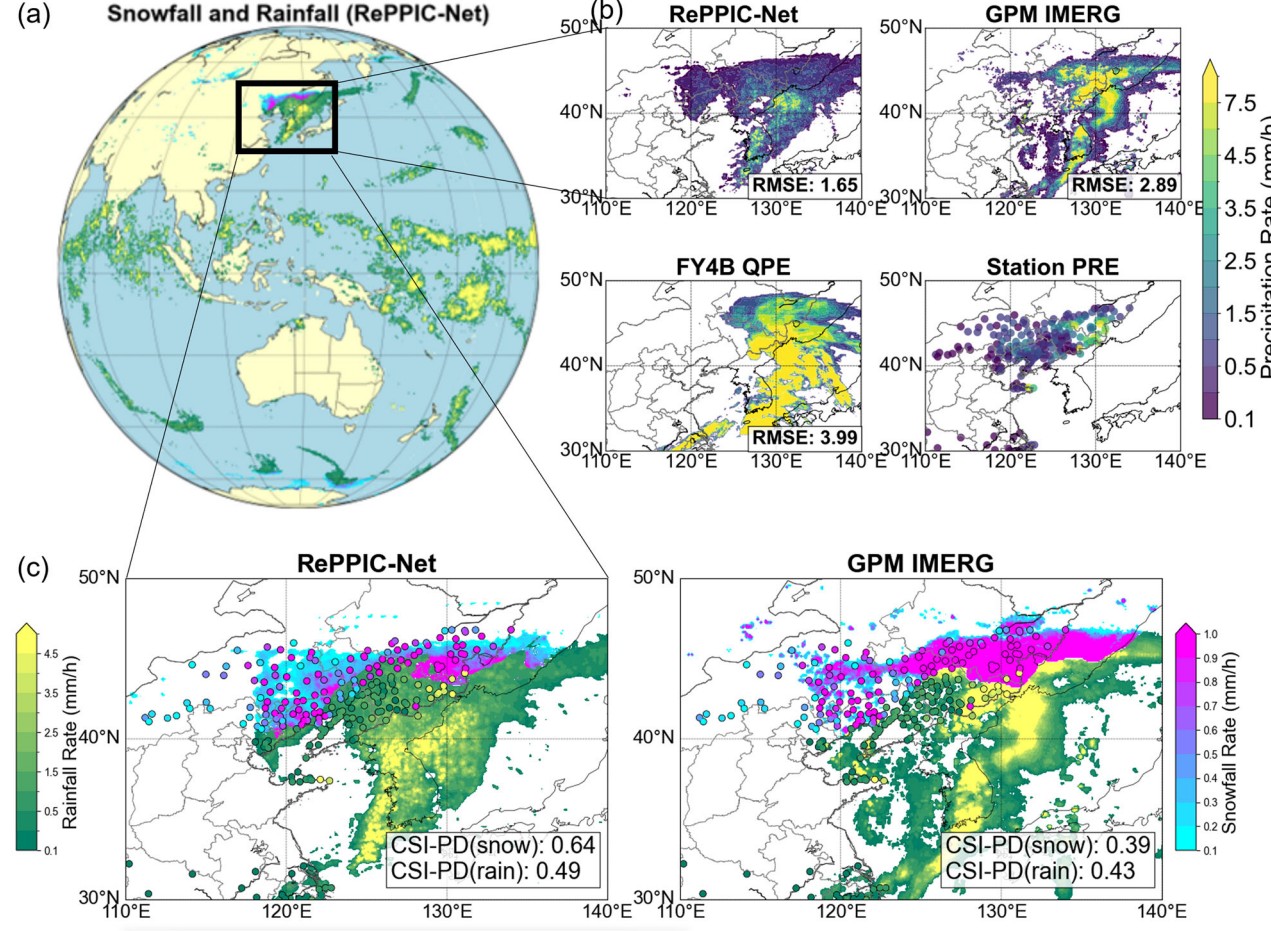

**Fig. 4 | Mixed-phase precipitation event observed on 5 November 2023 at 20:00 UTC. a, c** share the same colorbar to distinguish precipitation phase (snow/rain), while (**b**) displays only precipitation intensity without phase discrimination. Station observations are overlaid as circles in both (**b**) and (**c**) (outline: station location; filled color: observed precipitation rate).

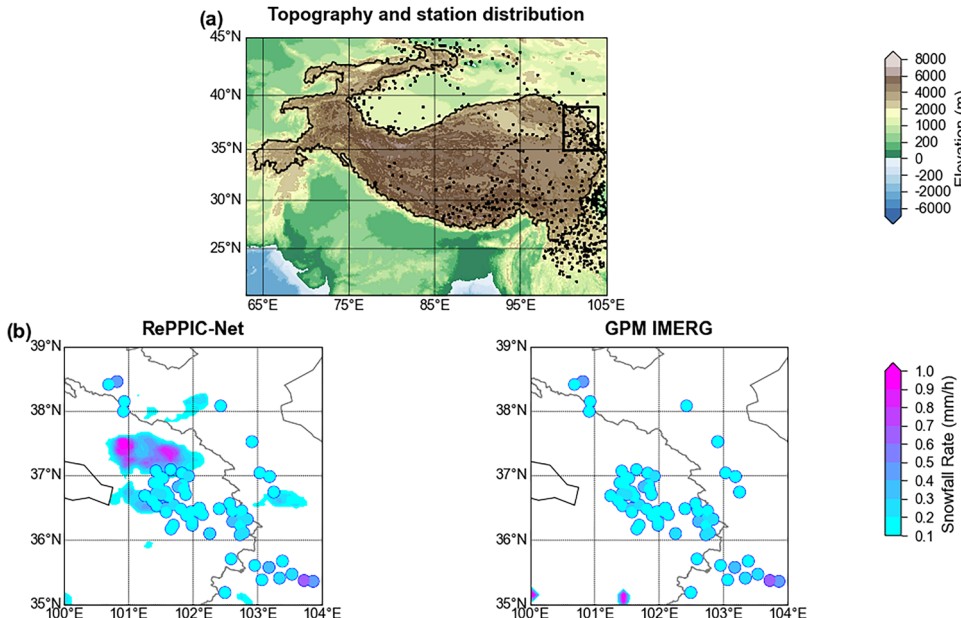

**Fig. 5 | Topographic context and a typical snowfall event over the Tibetan Plateau. a** Topographic map of the Tibetan Plateau and distribution of observation stations (black dots). The black rectangle marks the study area (100–104°E, 35–39°N). **b** A snowfall event over the Tibetan Plateau at 19:00 UTC on 14 December 2023. Station observations are overlaid as circles (outline: station location; filled color: observed precipitation rate). Gray lines denote administrative boundaries.

Figure 5 illustrates a snowfall event over the Tibetan Plateau (19:00 UTC on 14 December 2023), emphasizing challenges in solid precipitation monitoring over complex high-altitude terrain. GPM IMERG exhibited limited capability in detecting snowfall. RePPIC-Net achieved more precise localization of snowfall with enhanced spatio-temporal resolution (0.05° spatial, 15 min temporal) by integrating profiles of meteorological variables, while providing certain insights for snowfall monitoring in data-sparse regions lacking ground stations. However, the precipitation detection capabilities of satellite-based precipitation observations for complex plateau precipitation remain generally limited (Supplementary Fig. 1) and require further research.

## Performance of RePPIC-Net in nowcasting

Recent advances in geostationary satellite-based nowcasting have demonstrated the potential of data-driven models for cloud image forecasting. The DAYU model[37], leveraging Himawari-8 brightness temperature (BT) extrapolation, achieves correlation coefficients (CC) exceeding 0.9 within a 3-hour forecast window at 0.05° spatial resolution. Based on this framework, we adapted the architecture to process data from the FY-4B Advanced Geostationary Radiation Imager (AGRI) for brightness temperature (BT) forecasting. To evaluate the errors introduced by BT forecasts, we utilized 1, 2, and 3-hour leadtime satellite BT data as respective inputs to the precipitation model. Quantitative verification of rainfall revealed decreased CSI-PD metrics with lead times increasing from 0.29 (1 h) to 0.28 (2 h) and 0.27 (3 h), while snowfall metrics remained relatively stable at 0.15 (1 h), 0.15 (2 h), 0.14 (3 h). This stability in snowfall detection may reflect beneficial smoothing effects from motion field extrapolation, which enhanced hit rates without significantly increasing false alarms.

Building on the previous analysis, we now present nowcasting results for the representative case study shown in Fig. 4. The refined DaYu-FY model exhibits progressive decay in forecast sharpness with increasing leadtimes, showing moderate blurring at 3 h while retaining discernible cloud structure and BT patterns (Fig. 6a). When applied to precipitation nowcasting via RePPIC-Net using these forecasting FY-4B BT inputs, the system maintains robust identification of snowfall systems across the 3-hour window, with a similar CSI-PD score (0.37). However, a slight performance decline emerges for rainfall prediction

with the leadtimes increasing. To confirm this snowstorm case is representative, its heavy snowfall (>1 mm/h) CSI-PD scores (0.23 for both 1 h and 3 h forecasts) closely align with the 4-month systematic evaluation (more validation in Appendix), demonstrating consistent performance with the event given in Fig. 6. This divergence highlights RePPIC-Net's inherent resilience to input degradation—despite increasing BT field uncertainty and spatial smoothing, the model preserves critical skill in precipitation localization. The results underscore the framework's operational viability for short-term precipitation prediction while identifying key challenges for liquid-phase precipitation applications.

## Relationship between meteorology and precipitation phase

To understand the physical mechanisms of the model and the influence of meteorological fields on precipitation phase, we identified the most influential meteorological variables for our precipitation model via a random forest feature importance analysis (Tables 1 and 2). Following this identification, we analyzed the systematic errors in the 12 h FuXi forecast fields relative to ERA5 reanalysis data over China (Supplementary Table 1). Subsequently, we investigated the impact of perturbing these top-ranked variables on precipitation.

The relative importance of contributing features in precipitation detection was systematically evaluated through quantitative analysis (Table 1). Satellite channel differences emerged as highly significant predictors, as they effectively capture the essential changes in cloud properties and atmospheric conditions, which are fundamental for accurate precipitation detection. Among meteorological parameters, total precipitation shows the greatest impact on precipitation detection due to its direct reflection of the intensity and distribution of rainfall events. In addition, the relative humidity at 700 hPa and 500 hPa ranked prominently, highlighting the critical role of mid- to upper-tropospheric water vapor advection in modulating precipitation initiation. The relative importance of contributing features in quantitative precipitation estimation was also systematically evaluated (Table 2). Notably, cloud cluster characteristics—specifically the distance to cloud boundary and the minimum 10.8 μm infrared brightness temperature within clusters—were found to exhibit high importance rankings, suggesting that incorporating cloud cluster

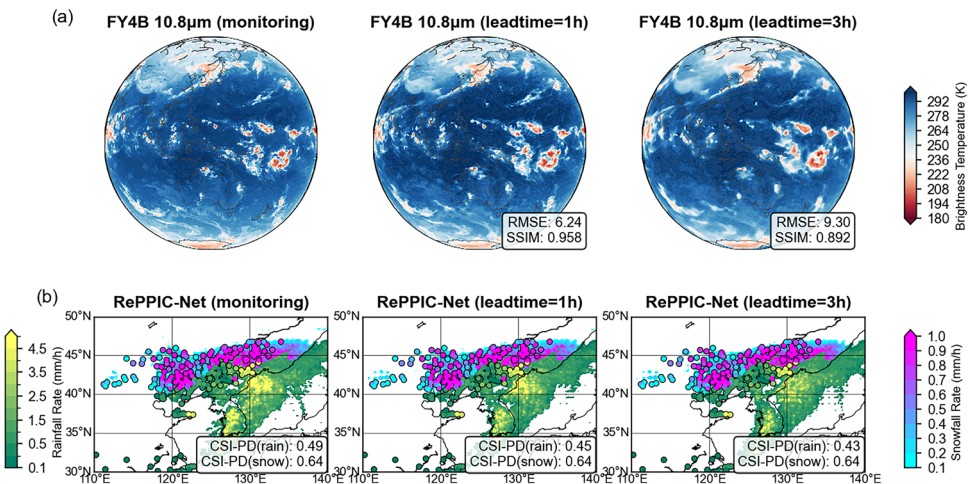

**Fig. 6 | Mixed-phase precipitation event nowcasting on 5 November 2023 at 20:00 UTC. a** FY-4B 10.8 μm brightness temperature; (**b**) RePPIC-Net precipitation nowcasting at 0, 1, 3 h leadtime.

**Table 1 | Importance scores of variables of Precipitation Detection and their corresponding rankings**

| Variables Score | Ranking | Variables Score | Ranking |
|---|---|---|---|
| $\Delta T_{6.25-10.8} = 0.072$ | 1 | $BT_{12.0} = 0.0352$ | 15 |
| $\Delta T_{7.42-12} = 0.072$ | 2 | R850 = 0.0328 | 16 |
| TP = 0.058 | 3 | $BT_{10.8} = 0.0328$ | 17 |
| R700 = 0.048 | 4 | Elevation = 0.0319 | 18 |
| $BT_{8.55} = 0.046$ | 5 | R925 = 0.0317 | 19 |
| R500 = 0.045 | 6 | $BT_{13.3} = 0.0313$ | 20 |
| $\Delta T_{10.8-12} = 0.044$ | 7 | Div500 = 0.0261 | 21 |
| Longitude = 0.040 | 8 | DIV850 = 0.0254 | 22 |
| T500 = 0.0373 | 9 | DIV700 = 0.0253 | 23 |
| T700 = 0.0366 | 10 | DIV925 = 0.0251 | 24 |
| T925 = 0.0365 | 11 | $BT_{7.42} = 0.0245$ | 25 |
| T850 = 0.0361 | 12 | $BT_{6.25} = 0.0222$ | 26 |
| SAZ = 0.0358 | 13 | $BT_{6.95} = 0.0218$ | 27 |
| Latitude = 0.0352 | 14 | – | – |

**Table 2 | Importance scores of variables of Quantitative Precipitation Estimation and their corresponding rankings**

| Variables Score | Ranking | Variables Score | Ranking |
|---|---|---|---|
| DIST2BDRY = 0.073 | 1 | Slope = 0.03200 | 18 |
| $BT_{10.8 \, min} = 0.0486$ | 2 | Longitude = 0.0300 | 19 |
| R500 = 0.0423 | 3 | $\Delta T_{10.8-12} = 0.0299$ | 20 |
| DIV925 = 0.0397 | 4 | T850 = 0.0298 | 21 |
| SAZ = 0.0375 | 5 | $DIST2BDRY = 0.0284$ | 22 |
| R700 = 0.0372 | 6 | $\Delta T_{6.95-10.8} = 0.0275$ | 23 |
| R850 = 0.0354 | 7 | $\Delta T_{7.42-12} = 0.0263$ | 24 |
| T500 = 0.0355 | 8 | $\Delta T_{6.25-10.8} = 0.0238$ | 25 |
| DIV500 = 0.0352 | 9 | $BT_{6.95} = 0.0195$ | 26 |
| TP = 0.0349 | 10 | $BT_{6.25} = 0.0190$ | 27 |
| R925 = 0.0347 | 11 | $BT_{13.3} = 0.0157$ | 28 |
| DIV700 = 0.0340 | 12 | $BT_{7.42} = 0.0152$ | 29 |
| DIV850 = 0.0333 | 13 | $BT_{8.55} = 0.0146$ | 30 |
| Elevation = 0.0330 | 14 | $BT_{10.8} = 0.0123$ | 31 |
| T925 = 0.0328 | 15 | $BT_{12.0} = 0.0116$ | 32 |
| T700 = 0.0327 | 16 | Area = 0.008 | 33 |
| Latitude = 0.0325 | 17 | – | – |

information enhances quantitative precipitation estimate accuracy. In addition, lower-level divergence and upper-level humidity parameters occupied prominent positions in the ranking hierarchy, indicating their critical roles in modulating precipitation intensity. Conversely, single-channel infrared satellite data consistently demonstrated lower predictive importance, highlighting the limited utility of isolated infrared spectral information for quantitative precipitation retrieval.

Supplementary Table 1 presents the error analysis in the 12-hour FuXi forecasts and ERA5 reanalysis for meteorological variables used in precipitation retrieval models. FuXi meteorological fields demonstrate high overall accuracy, with temperature estimates exhibiting a higher correlation coefficient than relative humidity. With the national automatic weather station network in China, we evaluated the discrepancies of FuXi and ERA5 from October to March, 2023–2025. We found a systematic overestimation over temperature (except at 500 hPa) and underestimation of relative humidity in FuXi (across all pressure levels).

The comparable performance of FuXi and ERA5 results in similar precipitation phase and detection (Supplementary Table 2). However, a slight systematic bias observed by FuXi leads to simultaneously lower POD and FAR of precipitation detection for snow and rain. This pattern reveals that the precipitation model has an inherent tendency to predict more precipitation events under the specific thermodynamic

profile of ERA5. This can be attributed to ERA5's generally lower temperatures (except at 500 hPa) and higher humidity at all levels—conditions that favor precipitation generation, particularly given the high feature importance of relative humidity at 700 hPa (R700) in our model. Furthermore, the ERA5-driven model exhibits a greater propensity to predict snow. This is likely due to its lower near-surface temperatures, which yield lower wet-bulb temperatures. The FuXi's slight warm bias appears to partially compensate for the inherent cool bias introduced by using pressure levels as a proxy for the actual surface conditions. Notably, inherent uncertainties in ERA5 within cloud systems may contribute to these differences[38].

## Discussion

Accurate real-time precipitation phase discrimination and quantification remain a critical unmet need for disaster mitigation and hydrological operations. Traditional numerical systems, constrained by coarse spatiotemporal resolution and computational latency, struggle to resolve rapidly evolving mesoscale phenomena such as blizzard transition events.

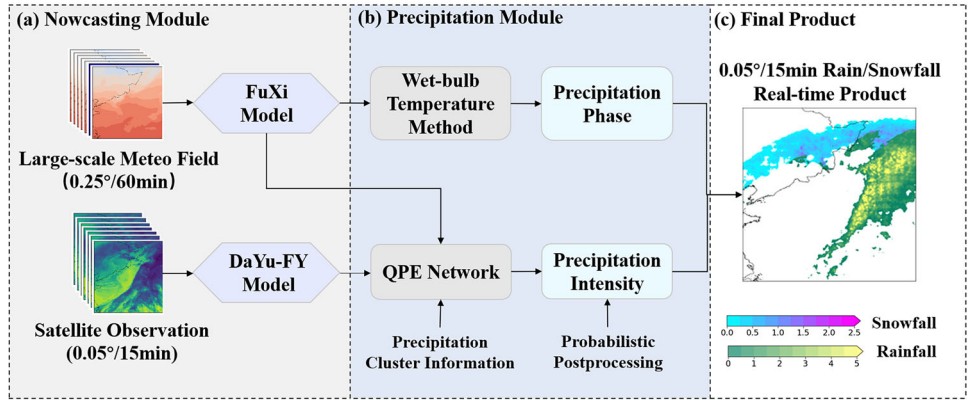

**Fig. 7 | Flowchart of the Real-time Precipitation Phase-Intensity Collaborative Retrieval Network (RePPIC-Net).** It comprises three core components: (**a**) Nowcasting Module, which combines 12-hour vertical atmospheric profiles (temperature, relative humidity) forecasts by the FuXi model (initialized with ERA5 reanalysis) with brightness temperature (Tb) data from either the DaYu-FY model (for precipitation nowcasting) or direct FY-4B satellite observation (for precipitation monitoring);(**b**) Precipitation Module, where the wet-bulb temperature (Tw) method is applied for phase classification (rain/snow) and a QPE network retrieves precipitation intensity; and (**c**) Final Products, which synthesizes the phase and intensity information to produce the final quantitative precipitation product with discriminated rain/snow phases.

Our work demonstrates that FuXi-satellite hybrid frameworks can transcend these limitations: RePPIC-Net, through synergistic integration of atmospheric prediction model (FuXi, calculated within seconds) and satellite-optimized deep learning, achieves real-time operational monitoring while delivering concurrent phase discrimination and intensity quantification−capabilities unattainable by current geostationary systems. By hierarchically fusing FuXi's 0.25° atmospheric forecasts (generated within seconds) with 0.05°/15 min satellite observations, RePPIC-Net reduces 4 h latency while achieving 3% (rainfall) and 57% (snowfall) improvements in CSI-PD over GPM IMERG, as validated across over 2000 stations in China. By bridging the gap between satellite capabilities and operational meteorology, this work establishes an operational solution for concurrent precipitation phase monitoring, directly addressing blizzard forecasting and aviation safety priorities. Its satellite-optimized feature fusion architecture enables robust performance in radar- and stations-sparse regions, providing a scalable template for full-disk nowcasting.

Future efforts should prioritize integrating cross-calibrated microwave constellations to enhance phase discrimination in cold-air damming scenarios and expanding adaptability to polar-orbiting sensor networks through physics-aware transfer learning. The demonstrated success of RePPIC-Net−where deep networks refine convective-scale features while conserving mesoscale dynamics resolved by the FuXi global prediction system−suggests broader applicability for compound hazard prediction in next-generation digital twin frameworks. We anticipate this work will catalyze international collaboration on AI-enhanced nowcasting systems, particularly for high-impact weather under climate change.

## Methods
### Real-time Precipitation Phase-Intensity Collaborative Retrieval Network (RePPIC-Net)
Effective real-time monitoring of precipitation phase (rain/snow) and intensity requires synergistic integration of meteorological dynamics and satellite observations. We present RePPIC-Net, a hybrid framework that unifies phase-resolved retrieval and error-optimized intensity retrieval through hierarchical fusion of large-scale meteorological fields and fine-scale satellite features. The framework leverages the computational efficiency of the FuXi global weather prediction model, initialized with ERA5 reanalysis data, to generate 12 h forecasts of vertical atmospheric profiles (e.g., temperature, relative humidity). The atmospheric variables provide critical thermodynamic context, including wet-bulb temperature (Tw) derived from DEM-adjusted

vertical interpolation, which replaces conventional air temperature for probabilistic liquid precipitation estimation.

Complementing this, a lightweight ResUNet architecture processes FY-4B AGRI Level-1 (L1) data to extract spatial features of precipitation clusters. Integrating near-instantaneous meteorological forecast outputs from the FuXi model (computationally generated within seconds) with multi-spectral satellite observations, RePPIC-Net achieves real-time full-disk quantitative precipitation estimates (QPE) at 0.05° spatial and 15 min temporal resolution (see Fig. 7). This real-time capability addresses the limitations of existing systems such as IMERG-Early, which rely on delayed JMA forecasts, and outperforms infrared-only methods for capturing heavy precipitation events.

Key innovations also include a probabilistic postprocessing layer that corrects systematic biases using joint occurrence-intensity distributions and a precipitation masking module that outputs precipitation occurrence probabilities alongside precipitation cluster intensity. By integrating probabilistic information, the framework achieves enhanced accuracy in intensity estimation.

### Precipitation phase discrimination module
The precipitation phase classification algorithm in RePPIC-Net employs a probabilistic framework adapted from the GPM IMERG probability of liquid precipitation (PLP) methodology. Central to this module is the estimation of wet-bulb temperature (Tw), a thermodynamic variable recognized in previous studies as the most robust indicator for precipitation phase discrimination[39,40]. Its established superiority over traditional approaches−including those based on dew point temperature, surface air temperature (Ta), and even certain machine learning techniques[41]. Unlike IMERG-Final, which calculates PLP from ERA5 reanalysis data[11], our system integrates 12 h forecasts of temperature (T) and relative humidity (RH) from the FuXi model.

Saturation vapor pressure is derived from FuXi's T and RH using the Magnus-Tetens formula, with vertical interpolation optimized through DEM-adjusted elevation matching to select the nearest atmospheric layer to the surface. This ensures topographic consistency in thermodynamic calculations. The wet-bulb temperature is then iteratively solved from the psychrometric equation, which leverages the calculated saturation vapor pressure. PLP is calculated as follows:

$$PLP = \frac{1}{1 + \exp\left(-b\left(T_w - a\right)\right)}, \tag{1}$$

where $T_w$ is the wet-bulb temperature in °C, and $a$ and $b$ are parameters from the fitting performed separately over ocean and land, as referenced in Algorithm theoretical basis document for IMERG Version 07[11].

## Quantitative precipitation estimation framework

The RePPIC-Net framework integrates three advancements to overcome limitations in geostationary satellite-based quantitative precipitation estimation (QPE). First, meteorological fields derived from the FuXi model replace satellite radar/microwave inputs, compensating for the constraints of infrared observations. Second, a ResUNet architecture incorporates six spatial features of precipitation clusters, including intra-cluster minimum brightness temperature ($BT_{10.8\,min}$), spectral differences ($BT_{12.3} - BT_{10.8}$), distance to boundary, distance to center, BT slope, and precipitation cluster area. Third, a Bayesian refinement layer reduces systematic biases by leveraging phase likelihoods and joint intensity distributions.

## Data preprocessing

FuXi forecast fields, FY-4B/AGRI L1 data, and geospatial ancillary datasets (e.g., DEM) are harmonized to a 0.05°/15 min resolution. Temperature and relative humidity profiles are vertically interpolated using DEM-adjusted elevations for Tw calculation, which serves as the driving variable for phase probability estimation. Precipitation labels are aligned with GPM DPR radar observations to ensure consistency during training and validation. The training dataset spans June 2022 to September 2023. To ensure data quality and model generalizability, rigorous preprocessing steps were implemented:

1. Invalid Data Filtering: Exclude samples with over 40% NaN values or precipitation slices with entirely zero values.
2. Zenith Angle Thresholding: Remove observations with zenith angles exceeding 65°, as larger angles degrade satellite measurement reliability due to increased atmospheric path length.
3. Label Truncation: Truncate precipitation intensity values at 100 mm/h to mitigate the influence of extreme outliers.
4. Normalization: Input features are normalized to zero mean and unit variance.
5. Sliding Window Sampling: Generate $32 \times 32$ pixels samples by sliding a window with a step size of 16, ensuring sufficient training while preserving spatial continuity. The $32 \times 32$ pixels input size aligns with typical convective-scale precipitation systems, capturing localized dynamics without excessive computational overhead.

## Deep learning model for precipitation detection and retrieval

We employed the UNet and ResUNet architectures for precipitation detection and retrieval tasks, respectively, as illustrated in Fig. 8. Residual Blocks are designed to learn residual mappings, which stabilize training and enhance feature reuse.

The UNet framework implements an encoder-decoder structure and skip connections to preserve multi-scale spatial features. The $32 \times 32$ pixels inputs (27 channels) are processed through three hierarchical encoding stages. Each encoder stage employs dual $3 \times 3$ convolutions (ReLU-activated) followed by $2 \times 2$ max pooling, progressively increasing feature channels ($64 \rightarrow 128 \rightarrow 256$). The bottleneck layer expands to 512 channels via two $3 \times 3$ convolutions. The decoder is upsampled using the transpose convolution, connecting the skip connections of the encoder outputs and refining the merged features by dual $3 \times 3$ convolutions. The final $1 \times 1$ convolutional layer with sigmoid activation generates $32 \times 32$ precipitation probability maps.

The ResUNet model integrates residual blocks within a UNet framework, enhancing feature propagation and mitigating gradient degradation. The encoder-decoder structure comprises four hierarchical levels, each containing two residual blocks with skip connections. Each residual block consists of sequential operations: a $3 \times 3$ convolutional layer, batch normalization, and ReLU activation. The encoder progressively downsamples spatial dimensions through max-pooling layers, while the decoder utilizes bilinear upsampling to restore resolution. The final layer applies a $1 \times 1$ convolution with sigmoid activation for precipitation probability estimation and linear activation for quantitative retrieval.

For model optimization, we employed the Adam optimizer with hyperparameters meticulously tuned as follows: the initial learning rate was set to $\alpha = 0.001$, balancing convergence stability and gradient descent efficiency; the batch size was configured at 16, a moderate choice that harmonizes computational tractability with gradient estimation reliability. For precipitation detection (a binary classification task with a 0.1 mm/h threshold), we utilized binary cross-entropy (BCE) loss to penalize probabilistic divergence between predicted class likelihoods and labels. For precipitation intensity retrieval, we employed a binary cross-entropy loss. This approach frames the retrieval of intensities exceeding a defined threshold as a probabilistic classification task, which demonstrated superior robustness to the inherent uncertainties in IR retrieval and the highly skewed distribution of precipitation values compared to regression losses (e.g., RMSE). The training epochs were task-specific: the detection model underwent 20 epochs with early stopping to prevent overfitting, while the retrieval model, requiring finer-grained parameter tuning for continuous target approximation, was trained for 150 epochs with periodic validation checks to monitor generalization performance. This dual-stage optimization framework ensures adaptive task-specific learning while maintaining algorithmic consistency, optimizing both discriminative accuracy for detection and parametric fidelity for retrieval. To align with the spatial resolution of GPM DPR precipitation data and minimize invalid values, the input and output dimensions were fixed to $32 \times 32$ pixels. This configuration balances computational efficiency and feature extraction capability for localized precipitation patterns. To validate the effectiveness of our approach, we conducted comprehensive comparisons against existing methods (random forests)[42,43] based on GPM DPR over the full-disk area. Our deep learning framework with meteorological field inputs achieved a CSI of 0.38 (at 0.1 mm/h threshold), significantly outperforming both point-to-point random forest approaches (0.31) and satellite-only deep learning configurations (0.33). Notably, even the random forest method augmented with meteorological fields reached only 0.36, underscoring the distinct advantage of our integrated architecture that combines area-based processing with atmospheric profile information.

## Deep learning model for satellite nowcasting

We propose a spatiotemporal Vision Transformer-based ResUNet architecture named DaYu-FY for infrared brightness temperature prediction. This model integrates 3D patch embedding, hierarchical attention (via Pyramid Vision Transformer, PVT), and convolutional encoder-decoder blocks.

The DaYu-FY adopts a hybrid encoder-decoder architecture, combining convolutional ResUNet modules with a Transformer-based attention mechanism. The input—a multi-frame, multi-channel spatiotemporal tensor—is initially processed by a cube embedded module, which partitions it into patch-wise tokens. The encoder employs a series of ResNet convolutional layers to extract deep spatial features, while incorporating temporal embeddings. These encoded representations are further processed by a Pyramid Vision Transformer (PVT), which utilizes multi-stage hierarchical attention to capture long-range spatial and temporal dependencies. The decoder path consists of ResNet modules that progressively reconstruct the output, leveraging skip connections from corresponding encoder layers to preserve spatial detail. A final linear projection head followed by reshaping restores the high-resolution prediction, yielding outputs that are spatially aligned with the original input—i.e., infrared brightness temperature predictions.

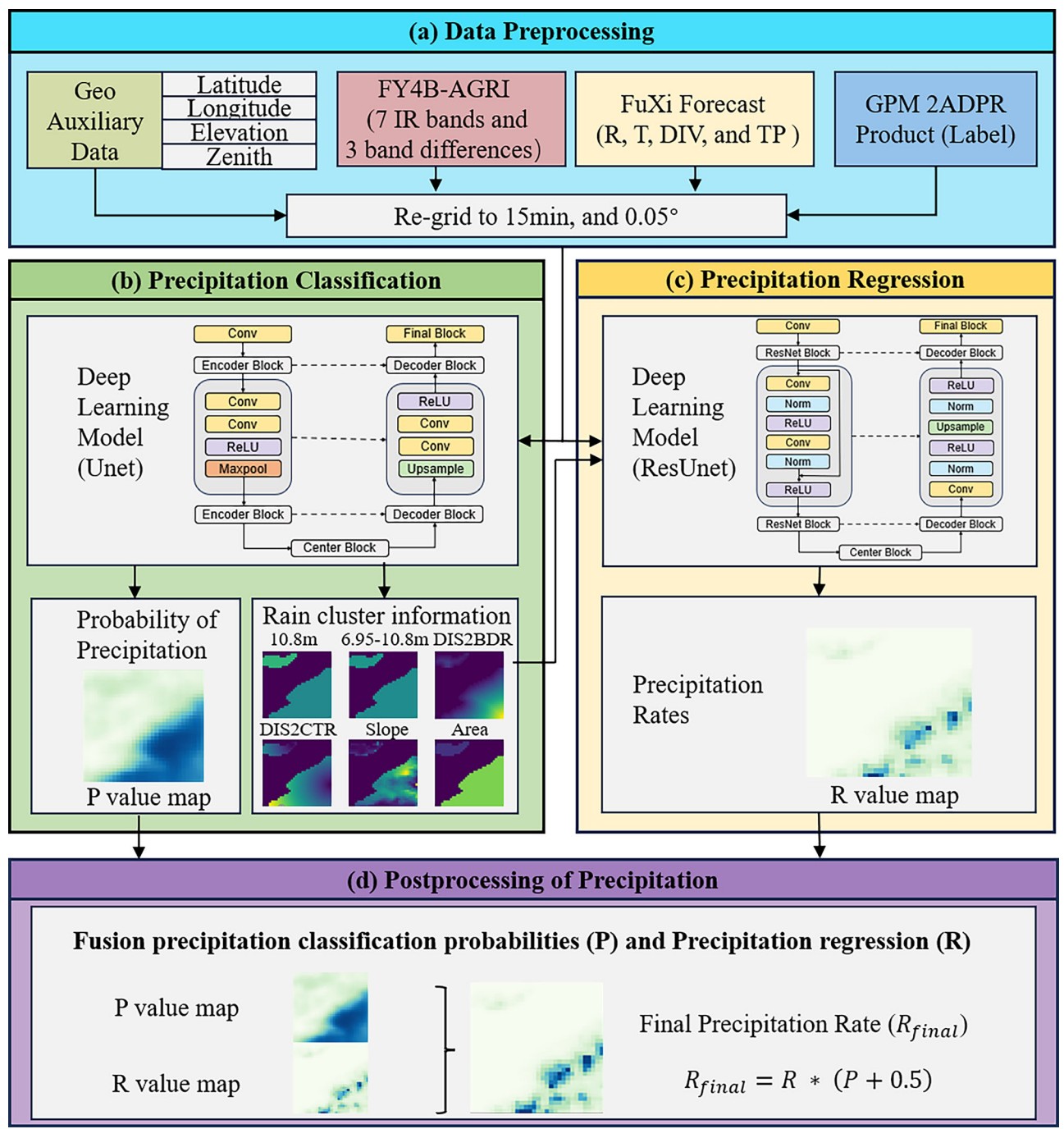

**Fig. 8 | Flowchart of the Precipitation Intensity Retrieval.** The procedure begins with (**a**) data preprocessing, where input datasets are spatially and temporally unified to a resolution of 0.05° and 15 min. The datasets are then fed into two deep-learning architectures: (**b**) a UNet for precipitation detection (classification), followed by (**c**) a ResUNet for intensity estimation (regression). Finally, (**d**) a post-processing step refines the output to produce the final precipitation rate.

For model optimization, we employed the AdamW optimizer with a carefully selected initial learning rate of $1 \times 10^{-5}$, promoting stable and gradual convergence, particularly suited to transformer-based models. The loss function is Smooth L1 loss with a small β value of 0.0001, which balances robustness to outliers with sensitivity to fine-grained prediction errors—an effective choice for regression tasks such as brightness temperature forecasting. Training was conducted over 10 epochs, a compact yet adequate schedule for convergence under constrained batch dynamics. To accommodate memory limitations and support high-resolution input, the batch size was set to 1. A Step LR scheduler was utilized with a step size of 1 epoch and a decay factor (γ)

of 0.3, allowing for rapid learning rate annealing and improved stability during the later stages of training.

### Implement details

All model training and inference were conducted on the Computing for the Future at Fudan (CFFF) high-performance computing platform. Each computing node was equipped with two AMD EPYC 7H12 64-core processors and one NVIDIA RTX A6000 GPU (50 GB memory). The software environment consisted of Ubuntu 22.04 LTS, Python 3.10, PyTorch 2.0.0, and CUDA 11.8. The models were trained using AdamW optimizers, with learning rates of $1 \times 10^{-4}$ for UNet/ResUNet and

$1 \times 10^{-5}$ for the DaYu-FY model. Batch sizes were set to 16 (for UNet/ResUNet) and 1 (for DaYu-FY). All experiments were performed under identical environments to ensure reproducibility. Supplementary Table 3 summarizes the computational complexity (FLOPs) and parameter counts (Parameters) of the major sub-models in the RePPIC-Net framework. The UNet and ResUNet modules are responsible for precipitation phase classification and quantitative precipitation retrieval, respectively. The FuXi short model generates real-time atmospheric physical fields, while the DaYu-FY model performs satellite BT extrapolation for precipitation nowcasting. Measured on an NVIDIA A6000 (50 GB) GPU setup, a single end-to-end full-disk (0.05°) inference combining precipitation phase classification and precipitation retrieval takes approximately 77.41 s; The subsequent PLP calculation for phase discrimination requires an additional 2 s. The runtime of the upstream forecasting models that supply inputs is: the FuXi short model takes about 5.1 s to produce a 6-hour forecast, and the DaYu-FY model takes about 2 s to produce a 30 min BT extrapolation. All timings are measured under the hardware/software environment stated in Methods and represent typical inference times for the reported resolution and product. Thus, a real-time forecast can be completed within 2 minutes.

Furthermore, we have provided a detailed explanation on the real-time operational capability of our product. The GPM IMERG-Early product, which requires the integration of multi-orbit and multi-sensor satellite observations and relies on meteorological fields from conventional numerical models, exhibits a latency of approximately 4 h. In contrast, although our model was trained and validated with ERA5 reanalysis data with a one-week delay, it can operationally utilize near-real-time ECMWF Operational Products (Ops), which have a latency of about 8 h. By incorporating the FuXi 12-hour forecast, meteorological fields covering approximately up to 4 h forecasts are obtained. This configuration enables our system to generate real-time precipitation monitoring and 1–3 h forecast products in approximately 2 min of computation.

## Evaluation settings

We conducted a comprehensive evaluation of the Real-time Precipitation Phase-Intensity Collaborative Retrieval Network (RePPIC-Net), leveraging multi-source observational data from the GPM IMERG-Late product and China's national ground stations (comprising over 2000 stations). The validation framework incorporated cross-comparisons with operational precipitation products, including full-disk precipitation intensity analysis against the FY-4B official precipitation products and systematic benchmarking using ground station observations against the GPM IMERG-Late dataset. Targeted assessments of precipitation intensity and phase classification accuracy focused on the low-intensity range (0.1–5 mm/h), as precise phase discrimination at these thresholds carries critical operational implications. This intensity band exhibits heightened sensitivity in meteorological impact differentiation–liquid precipitation within this range corresponds to light/moderate rainfall, while equivalent solid precipitation intensities may represent heavy snow or blizzard conditions.

Training and testing utilized FY-4B satellite observations from China's latest geostationary meteorological satellite, operational since June 2022. The model was trained on data from June 2022 to September 2023, with independent validation in a subsampled 11-month period (one day every three days) covering two winter seasons (October-March 2023-2024 and 2024-2025), February 2024 was excluded due to satellite orbital adjustments. All reported metrics reflect performance on the held-out test set. Model capabilities were rigorously assessed through: (a) precipitation phase discrimination using CSI, FAR, Frequency Bias (FB), Success Ratio (SR) and Probability of Detection (POD), and (b) precipitation intensity estimation evaluated via Root Mean Square Error (RMSE)[23]. Additionally, uncertainties were estimated following the method described in Jolliffe & Stephenson (2011)[44]. To holistically assess snowfall and rainfall monitoring performance, we introduced a composite metric, Critical Success Index for Phase and Detection (CSI-PD), defined as:

$$CSI - PD = P(Phase \cap Detection) = CSI_P \cdot CSI_D^n \qquad (2)$$

where: $CSI_P$ represents phase classification accuracy (snow/rain discrimination), $CSI_D^n$ denotes neighborhood-based precipitation detection skill, computed using a 0.5° spatial tolerance to mitigate spatiotemporal localization uncertainties, and CSI-PD quantifies the capability to simultaneously detect precipitation occurrence and correctly classify its phase.

This probabilistic formulation explicitly disentangles phase identification fidelity ($CSI_P$) from precipitation occurrence detection reliability ($CSI_D^n$). The joint optimization ensures balanced performance in both phase classification and detection tasks. The multiplicative combination enforces joint optimization, ensuring balanced performance across both tasks. The metric inherently penalizes scenarios where algorithms exhibit high precipitation detection rates but poor phase discrimination (or vice versa), precisely aligned with World Meteorological Organization (WMO) operational mandates for integrated precipitation assessment frameworks (WMO, 2023)[45].

## Datasets and baselines

**China ground station dataset.** We utilized high-quality hourly precipitation data from over 2000 national standard meteorological stations in China. These datasets were employed to evaluate the precipitation phase products of RePPIC-Net and conduct comparative analyses with the GPM IMERG-Late products. All data are accessible through the National Meteorological Information Center (NMIC).

**FY-4B dataset.** We leveraged two datasets from the Fengyun-4B (FY-4B) satellite's Advanced Geostationary Radiation Imager (AGRI): the full-disk L1 radiation observations and the Level-2 (L2) quantitative precipitation estimation product (FY-4B QPE-official)[46,47], both with a 4 km spatial resolution and a 15 min temporal resolution. We focused on the infrared spectral band from 6.25 to 13.3 μm to ensure continuous day-and-night precipitation monitoring. The FY-4B QPE-official product serves as the baseline for full-disk quantitative precipitation assessment. All FY-4B datasets, including observation data and related products, are available to the global community on the NSMC satellite data server website (http://satellite.nsmc.org.cn).

**GPM dataset.** The Global Precipitation Measurement (GPM) mission, a joint initiative of the National Aeronautics and Space Administration (NASA) and Japan Aerospace Exploration Agency (JAXA), aims to accurately measure precipitation on a global scale.

Dual-frequency Precipitation Radar (DPR), aboard the GPM Core Observatory satellite, is an active radar designed for observing precipitation and its vertical structure. Surface precipitation rates from normal scans (NS) were selected as labels for training precipitation estimation models[11,48].

The Global Precipitation Measurement (GPM) Integrated Multi-satellitE Retrievals for GPM Late Run (IMERG-Late) combines various infrared (IR) and microwave (MW) data to provide precipitation estimates at relatively high spatial (0.1°) and temporal (30 min) resolution, with a latency of 14 h[11]. IMERG-Late was selected as the reference dataset to evaluate the full-disk performance of the model product. These data are publicly available on the official website of the Goddard Earth Sciences Data and Information Services Center (GES DISC) (https://disc.gsfc.nasa.gov).

## DEM(GEBCO_2023)

The GEBCO_2023 Grid is a globally unified elevation model (15 arcsecond resolution), integrating terrestrial elevation and oceanic bathymetry, jointly developed under the Nippon Foundation-GEBCO Seabed

2030 Project (GEBCO, 2023). This authoritative dataset is publicly available at https://www.gebco.net (accessed 25 November 2023).

**Full-disk validation of different satellite precipitation products**
To quantify full-disk precipitation retrieval accuracy, we benchmarked RePPIC-Net and FY-4B's full-disk precipitation products against the multi-satellite GPM IMERG-Late dataset, a state-of-the-art reference for global precipitation estimation. Spatial error analysis (Supplementary Fig. 2) reveals that RePPIC-Net achieves comparable root-mean-square error (RMSE: 0.09) to FY-4B's operational precipitation product but exhibits distinct regional biases: systematic underestimation near Antarctica and overestimation over oceanic regions. In contrast, FY-4B demonstrates pronounced precipitation overestimation at high southern latitudes and across the disk's eastern quadrant—a pattern likely associated with increased atmospheric path length and signal attenuation at oblique viewing angles, which degrade radiometric fidelity.

When evaluated against GPM IMERG's surface precipitation rates and phase classifications (Supplementary Table 4), RePPIC-Net outperforms FY-4B in both rainfall and snowfall retrievals, particularly reducing the RMSE for snowfall by 58% (0.19 vs. 0.46). This performance gap persists despite excluding phase discrimination capabilities, underscoring RePPIC-Net's superior capacity to resolve precipitation intensity across hydrometeor types. These results position RePPIC-Net as a robust framework for full-disk precipitation retrieval, especially for snowfall estimation.

## Data availability

The FY-4B satellite datasets were obtained from the National Satellite Meteorological Center (NSMC) satellite data server (http://satellite.nsmc.org.cn). The GPM datasets were accessed from the Goddard Earth Sciences Data and Information Services Center (GES DISC, https://disc.gsfc.nasa.gov), specifically GPM DPR via https://data.nasa.gov/dataset/gpm-dpr-precipitation-profile-l2a-1-5-hours-5-km-v07-gpm-2adpr-at-ges-disc-579a6 and GPM IMERG Late via https://data.nasa.gov/dataset/gpm-imerg-late-precipitation-l3-half-hourly-0-1-degree-x-0-1-degree-v07-gpm-3imerghhl-at-g-9e38a. The GEBCO_2023 dataset was downloaded from the General Bathymetric Chart of the Oceans (GEBCO, https://www.gebco.net/data-products/gridded-bathymetry-data/gebco2023-grid). To facilitate the verification of the model results presented in this study, the key precipitation data products have been uploaded to the Zenodo repository (https://zenodo.org/records/18376445). The station data that support the validation of the models for the China precipitation phase and intensity are available from the China Meteorological Administration, but restrictions apply to the availability of these data, which were used under license for the current study and so are not publicly available. Data can be accessed with the approval of the China Meteorological Administration via a formal request.

## Code availability

We rely on PyTorch (https://pytorch.org) for model training and cartopy (https://scitools.org.uk/cartopy) for geospatial data processing. The code of RePPIC-Net is available on https://doi.org/10.5281/zenodo.18357366.

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

## Acknowledgements

This work was supported by the National Natural Science Foundation of China (Grants 42450254 and 42222506 to Z.F., and 42475095 to L.H.), and by the Shandong Provincial Natural Science Foundation (Grant ZR2025LQX001 to L.H.). The computations were performed using the Computing for the Future at Fudan (CFFF) platform.

## Author contributions

Y.C., L.H., and Z.F. conceived and led the research project. L.H. and Z.F. explored and devised the methodology. Y.C. developed the core algorithms and wrote the original manuscript. Z.R. developed the satellite-based nowcasting methodology. L.H. collected and processed observational datasets. W.Y. supervised the research and revised the manuscript. G.M. coordinated project administration and resources. J.G. designed the meteorologist evaluation protocols and conducted expert assessments. Y.C. and Z.F. conducted experiments and analyzed results. Z.F. provided critical funding support for this work. Z.R. and T.X. provided a critical review to revise the manuscript. All authors reviewed and approved the final manuscript.

## Competing interests

The authors declare no competing interests.
