## [Transparent Peer Review file · Nature Communications]

Snow or Rain? Hybrid AI Deciphers Surface Precipitation Phase from Satellite Observations

Corresponding Author: Professor Feng Zhang

Version 0:

Reviewer comments:

Reviewer #1

(Remarks to the Author)

Review of manuscript "Snow or Rain? Hybrid AI Deciphers Surface Precipitation from Satellite" submitted to Nature Communications

General Comments

This manuscript deals with a new methodology to nowcast surface precipitation phase based on a hybrid AI framework. Overall the topic, development, and results are well suited for this journal. However, a number of clarifications and, mostly formal, corrections are needed to be considered acceptable for publication. I cannot recommend to publish it in its current form and I recommend a major review. Please see below a list of specific comments.

Specific Comments

1. Page 1, line 1. Suggest: Precipitation -> Precipitation Phase
2. Page 1, line 19. Please check English: avalanche -> avalanches
3. Page 2, line 22 (and 25). Please check English: phases -> phase [typically singular is used in this type of sentences]
4. Page 2, line 31. Please check English: ever-first -> first-ever
5. Page 2, line 31-32. Please rephrase: aiding the need
6. Page 2, line 38. Suggest rephrasing: transient snow-to-rain transitions
7. Page 5, lines 98-99. Are temperature and relative humidity 12-hour forecast only at surface (or near surface) level? It is not clear why below 'vertical atmospheric profiles' are mentioned – these profiles could indeed include temperature and relative humidity. Please rewrite to improve to clarify this sentence.
8. Page 5, line 100. Authors identify here the term T_{wet} with wet-bulb temperature, but later in the text T_w is used instead. Please be consistent.
9. Page 8. Is the CSI-PD formula equivalent to compute the CSI for phase type whenever precipitation is detected? Please clarify in text.
10. Page 8, lines 151-152: "where: $\langle cr \rangle CSI_p$ " -> "where CSI_p " [remove colon and carriage return]
11. Page 8, line 161. Could authors please add a reference regarding the WMO mandate mentioned?

12. Page 9, first paragraph. Some verification score values are listed here with 3 decimal digits, unlike in other parts of the text. I suggest always using the same number of decimal digits and also to include uncertainties for each score value – computation formulas for verification score uncertainties can be found, for example, in Jolliffe & Stephenson (2011). A summary table showing both scores and uncertainties can be included in this section, or in an appendix. Complementary error bars to some plots such as Fig. 2b.
13. Page 10, Fig. 2b. Please be consistent with notation of variables: for example, CSI-P was introduced in line 150 with p as a subscript – similarly for others. Please check and correct.
14. Page 11, lines 206-208. Please add the year of the dates mentioned.
15. Page 12, Fig. 3 (and Fig. 7 too). Typo (twice in Fig. 3): IMERGE.
16. Page 12, Fig. 3. Colour bar title: Precipitation [mm/h]; is it Precipitation rate or Rainfall rate? Similarly, Snowfall, shouldn't be Snowfall rate?
17. Page 13, lines 239-240. Please check 'integrating vertical meteorological fields' do you mean 'integrating profiles of meteorological variables'?
18. Page 14, Fig. 4 caption. Please add a full stop.
19. Page 21. To better connect the current study with existing literature, either in this section or elsewhere, authors could mention other studies describing methodologies to classify surface precipitation phase – examples could be, for instance, Casellas et al. (2021), Gao et al. (2024) or Øydvin et al. (2025) – of course others are possible. Additionally, authors should also comment, at least briefly, the recent article by Jennings et al. (2025) which is closely related with the manuscript submitted.
20. Page 23, first paragraph. The validation period (Oct-2023 to Jan-2024) did present comparable characteristics to the training period (Jun-2022 to Sep-2023)? Please comment and if necessary add a brief justification in the text.
21. Page 25, line 481. Please indicate here at which rainfall rate (or rates) was set the precipitation detection threshold.
22. Page 30, line 598. Fig. 9 not found.
23. Page 32, line 637. Is the 'monitoring evaluation period' the 'validation' period? Please use the same terms to avoid possible confusions.
24. Page 32, line 650 (and 652, 654-655). Please check English: were downloaded and downloadable?
25. Page 36, Fig. 6. Please add in the figure caption the IMERG version (Late?) as in Fig. 8.
26. Page 37, Fig. 7 caption. The content of panels a) and b) should be indicated.

References

- Casellas, E., et al. (2021). Nowcasting the precipitation phase combining weather radar data, surface observations, and NWP model forecasts. *Quarterly Journal of the Royal Meteorological Society*, 147(739), 3135-3153.
- Gao, Y., et al. (2024). Performance of FY-4B GIIRS temperature products under cloudy skies and their enhancement of surface precipitation type forecasting. *Atmospheric Research*, 302, 107305.
- Jennings, K. S., et al. (2025). Machine learning shows a limit to rain-snow partitioning accuracy when using near-surface meteorology. *Nature Communications*, 16(1), 2929.
- Jolliffe, I. T., & Stephenson, D. B. (Eds.). (2011). *Forecast verification: a practitioner's guide in atmospheric science*. John Wiley & Sons.
- Øydvin, E., et al. (2025). Combining commercial microwave links and weather radar for classification of dry snow and rainfall. *Atmospheric Measurement Techniques*, 18(10), 2279-2293.

(Remarks on code availability)

Reviewer #2

(Remarks to the Author)

The motivation of the work is highly significant operational meteorology, hydrology, and disaster management because

Real-time, accurate precipitation phase distinction is a long-standing global challenge, especially in data-sparse mountainous and high-latitude regions where hazards like avalanches and blizzards are common.

The paper proposes an interesting combination of AI architectures, fusing physics-informed meteorological fields from a rapid AI weather model (FuXi) with high-resolution satellite observations and a nowcasting model.

The main results in the paper are that the authors want to develop a system for real-time operations that achieves comparable rain-detection performance between their model and IMERG-Late with CSI scores of 0.95 and 0.96 and similar bias in snowfall overestimation and rainfall underestimation. In a given case study, the authors demonstrate the superior skill of the model in predicting a high impact snow blizzard, outperforming IMERG and FY-4B QPE product.

Major Critical Comments:

- (1) The proposed training dataset seems quite small, training over June 2022 to September 2023, which covers likely only one snowfall season and one rainfall season, so it's unclear if there was any overfitting of the model.
- (2) When looking at the evaluation time period of October 2023 to January 2024, that again seems very small to draw any concrete conclusions from. It covers only the snowfall period and even then extending the evaluation period to include up until March 2024 would be better.
- (3) The evaluation should also cover the rainfall season so that we can understand any False alarm ratios better in cases when there is no rainfall as the model could be overfitting.
- (4) The paper claims to be highly performant for extreme precipitation events but only shows CSI for values up to 5mm/hr and only showcases one case study for a high impact event. From this case study it's really hard to draw any conclusions that the system is indeed highly performant over different scenarios.
- (5) Clarification of the "Real-Time" Capability: The central claim of the paper is its "real-time" operational capability, which is highlighted as a key advantage over systems with a "4-hour latency". However, the methods section states that the FuXi model is "initialized with ERA5 data (about 8-hour latency)". This creates a significant contradiction. While the FuXi model's forecast generation is computationally fast ("within seconds"), its dependency on an 8-hour-old initial state seems to undermine the "real-time monitoring" claim. The authors must clarify this point. Is the system truly real-time, or is it more accurately described as a rapid-update system with an 8-hour data lag? A detailed discussion of how this latency in the input data affects the final product's timeliness and accuracy is essential for the paper's credibility.
- (6) The paper claims accuracy at 0.05 degree spatial resolution but all results are against IMERG data which is at 0.01 degree resolution, so it's unclear what the validation mechanism is for accuracy at 0.05 degrees.
- (7) The authors introduce a novel and useful composite metric, the Critical Success Index for Phase and Detection (CSI-PD), defined as $CSIP \cdot CSIDn$. While the rationale is explained as penalizing models that are good at one task but not the other, the specific choice of multiplication versus other methods of combination (e.g., a weighted average) is not justified. A brief sentence explaining why this multiplicative, probabilistic formulation was chosen would strengthen the methodological contribution.

(Remarks on code availability)

Reviewer #3

(Remarks to the Author)

Please refer to the attached document

[Editorial Note: See end of file]

(Remarks on code availability)

The code appears to be well-written and free of technical issues.

Version 1:

Reviewer comments:

Reviewer #1

(Remarks to the Author)

I thank authors for the corrected version and the detailed item-by-item response provided. I think the new corrected version fulfilled all items raised so it can now be accepted for publication.

(Remarks on code availability)

Reviewer #2

(Remarks to the Author)

Thanks for providing detailed responses to the reviewer comments and updating the manuscript accordingly. Based on the

updates, the paper seems to have been significantly strengthened with robust results. The authors have effectively addressed the critical concerns regarding dataset size, operational latency, and physical interpretability.

The paper introduces RePPIC-Net, a hybrid AI framework that fuses 3D atmospheric physics from the FuXi model with geostationary satellite observations to quantify surface precipitation phase. This system represents a "first-ever" achievement in satellite-based nowcasting, offering a scalable template for global early warning systems.

The following key enhancements and justifications characterize this final version:

1. The authors have convincingly resolved concerns regarding operational latency. By initializing the FuXi model with ECMWF Operational Products, the system bridges data lags to generate 0–3 hour nowcasts with a total computational time of approximately 2 minutes.
2. The evaluation framework was significantly expanded to an 11-month period covering two complete winter seasons (2023–2025). Accuracy at the 0.05° spatial resolution was independently validated against more than 2,000 national standard meteorological stations in China.

I recommend an acceptance of this work based on its impact during wintertime in data-sparse regions, and operational qualities.

(Remarks on code availability)

The code mainly provides the ML architecture which may be useful for others to derive their model training methodologies from it.

Response Letter

Dear Reviewers,

We sincerely thank you for your insightful comments and valuable suggestions on our manuscript NCOMMS-25-50501A (originally titled “Snow or Rain? Hybrid AI Deciphers Surface Precipitation from Satellite”). In response to your feedback, we have comprehensively revised and improved the manuscript, with the following key enhancements:

Firstly, following your advice, we have refined the title to “Snow or Rain? Hybrid AI Deciphers Surface Precipitation Phase from Satellite” to better focus the scope of the study. Then, we have significantly **extended the evaluation period** (from October–March across 2023–2025) to ensure adequate representation of snowfall events, thereby enabling more robust assessment. We have also **optimized the evaluation metrics and strengthened the physical interpretation** section to provide more comprehensive analysis and insights. Moreover, we have **clarified the real-time applicability of our model**: ERA5 reanalysis data was used during model training and validation, near-real-time ECMWF operational analysis products with an 8-hour latency can be employed as meteorological inputs in operational applications. Our results demonstrate that the proposed RePPIC-Net model can generate the first phase-discriminable 1–3 hour precipitation forecast within 2 minutes, addressing a critical gap in operational forecasting.

In particular, to strengthen the rigor of our **evaluation and enhance physical interpretability**, we have implemented the following three major improvements:

1. **Quantified uncertainties** in precipitation phase identification and detection, and added a **new section (“The spatiotemporal error distributions”)** to systematically analyze the error distribution of the model;
2. Included a **new section (“Relationship between meteorology and precipitation phase”)** to enhance the physical interpretability of the model and analyze the physical mechanisms by which meteorological variables trigger precipitation;
3. **Added comparisons** with state-of-the-art algorithms to highlight the novelty of our approach, and quantified the performance gains achieved by incorporating meteorological inputs.

We believe that the revisions made in response to your expert suggestions have significantly enhanced the reliability and depth of the study's conclusions, and we kindly request your re-evaluation of the revised manuscript. In this response letter, **our response is presented in blue**, and any direct quotations from the revised manuscript are underlined.

The authors are thankful to the reviewers for their valuable comments. Here is our detailed point-by-point responses:

Reviewer 1:

General Comments

This manuscript deals with a new methodology to nowcast surface precipitation phase based on a hybrid AI framework. Overall the topic, development, and results are well suited for this journal. However, a number of clarifications and, mostly formal, corrections are needed to be considered acceptable for publication. I cannot recommend to publish it in its current form and I recommend a major review. Please see below a list of specific comments.

Specific Comments

1. Page 1, line 1. Suggest: Precipitation -> Precipitation Phase

Response: Thank you for your valuable suggestion. We have revised the article title accordingly to make it more focused. The revised manuscript title is:

"Snow or Rain? Hybrid AI Deciphers Surface Precipitation Phase from Satellite"

2. Page 1, line 19. Please check English: avalanche -> avalanches

3. Page 2, line 22 (and 25). Please check English: phases -> phase [typically singular is used in this type of sentences]

4. Page 2, line 31. Please check English: ever-first -> first-ever

Response: We greatly appreciate the reviewer's efforts in standardizing the grammar. All the suggested spelling and grammatical issues have been carefully revised accordingly.

5. Page 2, line 31-32. Please rephrase: aiding the need

6. Page 2, line 38. Suggest rephrasing: transient snow-to-rain transitions

Response: We thank the reviewer for the comment and have revised the phrase, changing "aiding the need" to "meeting the need". Additionally, "transient snow-to-rain transitions" has been modified to "transient phase shifts from snow to rain".

7. Page 5, lines 98-99. Are temperature and relative humidity 12-hour forecast only at surface (or near surface) level? It is not clear why below 'vertical atmospheric profiles' are mentioned – these profiles could indeed include temperature and relative humidity. Please rewrite to improve to clarify this sentence.

Response: Thank you for pointing out this issue. We have revised the text to clarify this aspect. The FuXi model generates 12-hour forecasts of temperature, relative humidity and wind fields (u and v) across 13 vertical pressure levels, not just at the surface level. These multi-level atmospheric profiles serve two critical purposes in our framework:

1). They provide essential three-dimensional atmospheric information as inputs to our precipitation retrieval model, complementing the limitations of infrared satellite channels alone.

2). We specifically select near-surface pressure levels based on digital elevation model (DEM) data to calculate wet-bulb temperature for precipitation phase discrimination.

The original statement has been modified to (page 5, lines 100-103):

“The framework leverages the computational efficiency of the FuXi global weather prediction model, initialized with ERA5 reanalysis data, to generate 12-hour forecasts of vertical atmospheric profiles (e.g., temperature, relative humidity).”

8. Page 5, line 100. Authors identify here the term T_{wet} with wet-bulb temperature, but later in the text T_w is used instead. Please be consistent.

Response: Thank you for pointing this out. We have uniformly used T_w to represent wet-bulb temperature throughout the text.

9. Page 8. Is the CSI-PD formula equivalent to compute the CSI for phase type whenever precipitation is detected? Please clarify in text.

Response: Thank you for pointing out the need for clarification. In fact, the CSI-PD formula is not equivalent to computing the CSI for phase type only when precipitation is detected. **CSI-PD quantifies the capability to simultaneously detect precipitation occurrence and correctly classify its phase.** It evaluates performance across all samples, including those where precipitation may not have been detected at all. Its formulation as $CSI_P \cdot CSI_D^n$ explicitly accounts for skill in both detection and phase discrimination. We have added a sentence in the corresponding section of the manuscript to clarify the meaning of CSI-PD. The modification is as follows (page 8, lines 160-167):

“……, and CSI-PD quantifies the capability to simultaneously detect precipitation occurrence and correctly classify its phase.

This probabilistic formulation explicitly disentangles phase identification fidelity (CSI_P) from precipitation occurrence detection reliability (CSI_D^n). The joint optimization ensures balanced performance in both phase classification and detection tasks. The

multiplicative combination enforces joint optimization, ensuring balanced performance across both tasks.

10. Page 8, lines 151-152: “where: <cr> CSI_p” -> “where CSI_p” [remove colon and carriage return]

Response: Thank you for your suggestion. The corresponding modifications have been made.

11. Page 8, line 161. Could authors please add a reference regarding the WMO mandate mentioned?

Response: Thank you for your suggestion. The relevant references have been added as follows (page 40, lines 765-766):

“World Meteorological Organization. Abridged Final Report of the Nineteenth World Meteorological Congress. WMO-No. 1326 (2023).”

12. Page 9, first paragraph. Some verification score values are listed here with 3 decimal digits, unlike in other parts of the text. I suggest always using the same number of decimal digits and also to include uncertainties for each score value – computation formulas for verification score uncertainties can be found, for example, in Jolliffe & Stephenson (2011). A summary table showing both scores and uncertainties can be included in this section, or in an appendix. Complementary error bars to some plots such as Fig. 2b.

Response: Thank you for pointing this out. We have **performed an uncertainty assessment of the verification scores using the methods from the recommended paper** (Jolliffe & Stephenson, 2011). Accordingly, we have updated Fig. 2(b) by replacing it with a summary table that includes all score values together with their corresponding uncertainties. To accurately represent these small uncertainty values, the data in the table are presented with four decimal places, whereas the figure retains two decimal places for visual clarity and easier comparison.

		GPM IMERG		RePPIC-Net	
		CSI	Uncertainty(±)	CSI	Uncertainty(±)
Phase	Rain	0.8252	0.0004	0.8477	0.0004
(CSI _p)	Snow	0.7099	0.0005	0.6896	0.0007
Detection	Rain	0.3713	0.0017	0.3713	0.0018

(CSI _D)	Snow	0.1411	0.0034	0.2283	0.0041
Overall	Rain	0.3064	0.0014	0.3147	0.0015
(CSI-PD)	Snow	0.1001	0.0024	0.1574	0.0028

Fig. 2 (b) CSI_P, CSI_D, CSI-PD and uncertainty of RePPIC-Net and GPM IMERG within the 0.1–5 mm/h precipitation intensity threshold.

13. Page 10, Fig. 2b. Please be consistent with notation of variables: for example, CSI-P was introduced in line 150 with p as a subscript – similarly for others. Please check and correct.

Response: We thank the reviewer for pointing this out. We have now consistently used the subscript notation (e.g., CSI_P) throughout the manuscript, including in Fig. 2b.

14. Page 11, lines 206-208. Please add the year of the dates mentioned.

Response: We thank the reviewer for this helpful suggestion. We have now added the missing years to the dates mentioned on page 14, lines 248–250.

“Under the influence of a strong cold air mass, a high-impact mixed precipitation event occurred across northeastern China from 20:00 UTC on 5 November 2023 to 20:00 UTC on 7 November 2023.”

15. Page 12, Fig. 3 (and Fig. 7 too). Typo (twice in Fig. 3): IMERGE.

Response: We sincerely appreciate the reviewer’s meticulous attention to detail. Following the identification of the typo "IMERGE", we have thoroughly revised both Fig. 3 and Fig. 6.

16. Page 12, Fig. 3. Colour bar title: Precipitation [mm/h]; is it Precipitation rate or Rainfall rate? Similarly, Snowfall, shouldn’t be Snowfall rate?

Response: We thank the reviewer for this insightful comment. The color bar title "Precipitation [mm/h]" has been clarified to "Precipitation rate (mm/h)", as the FY4B QPE product does not distinguish precipitation phase (thus this figure represents combined rain and snow). Similarly, "Snowfall" has been adjusted to "Snowfall rate" for consistency. To enhance clarity, we also have revised the color bar labels in Fig. 6 and Fig. 7, as well as the figure caption.

Fig. 5 | Mixed-phase precipitation event observed on 5 November 2023 at 20:00 UTC, (a) and (c) share the same colorbar to distinguish precipitation phase (snow/rain), while (b) displays only precipitation intensity without phase discrimination.

17. Page 13, lines 239-240. Please check ‘integrating vertical meteorological fields’ do you mean ‘integrating profiles of meteorological variables’?

Response: We appreciate the reviewer’s suggestion. We have revised the phrase “integrating vertical meteorological fields” to “integrating profiles of meteorological variables” in the original text to ensure clarity and precision.

18. Page 14, Fig. 4 caption. Please add a full stop.

Response: We appreciate the reviewer’s attention to detail regarding Fig. 4’s caption punctuation. While Fig. 4 already included a full stop, we conduct a review of all figure captions. We have now corrected this omission, ensuring all figure captions adhere to consistent academic formatting standards.

19. Page 21. To better connect the current study with existing literature, either in this section or elsewhere, authors could mention other studies describing methodologies to classify surface precipitation phase – examples could be, for instance, Casellas et al. (2021), Gao et al. (2024) or Øydvin et al. (2025) – of course others are possible. Additionally, authors should also comment, at least briefly, the recent article by Jennings et al. (2025) which is closely related with the manuscript submitted.

Response:

Thank you for this valuable suggestion. We have incorporated the recommended paper into our manuscript to better connect our methodology within the existing research. Specifically, we have:

1). Add references to Casellas et al. (2021), Gao et al. (2024), and Øydvin et al. (2025) in the introduction section reviewing precipitation phase discrimination methods. These citations have been integrated into the following context: "While active radars and microwave sensors on polar-orbiting satellites can detect precipitation phase transitions by analyzing vertical hydrometeor profiles through reflectivity variations, current satellite-based phase retrieval still demonstrates significant bias compared to NWP-based phase results²²⁻²⁶." (page 4, lines 64-67)

2). Add a discussion of Jennings et al. (2025) in the methodology section, as their recent work provides important insights into phase classification that closely align with our methodology. The revised text now reads (pages 21-22, lines 376-381): "Central to this module is the estimation of wet-bulb temperature (T_w), a thermodynamic variable recognized in previous studies as the most robust indicator for precipitation phase discrimination³⁹⁻⁴⁰. Its established superiority over traditional approaches—including those based on dew point temperature, surface air temperature (T_a), and even certain machine learning techniques⁴¹."

20. Page 23, first paragraph. The validation period (Oct-2023 to Jan-2024) did present comparable characteristics to the training period (Jun-2022 to Sep-2023)? Please comment and if necessary, add a brief justification in the text.

Response: We sincerely thank you for raising this important point. We agree that the original shorter validation period (October 2023 to January 2024) may have limitations in robustly representing the model's performance with the training period (June 2022 to September 2023). To thoroughly address this concern, we have **extended the validation period to 11 months, spanning two complete winters** (covering available months from October to March across 2023–2024 and 2024–2025; February 2024 was excluded due to satellite orbit adjustments and lack of available data). This optimized validation strategy not only increases the temporal duration but also introduces entirely new temporal sequences subsequent to the training period. This allows for a more rigorous evaluation of the model's generalization capability and provides a more reliable demonstration of its performance comparability with the training period. We have added a brief justification for this modification in the manuscript (page 8, lines 144-148) and have updated all corresponding evaluation figures.

"The model was trained on data from June 2022 to September 2023, with independent

validation in a subsampled 11-month period (one day every three days) covering two winter seasons (October-March 2023-2024 and 2024-2025), February 2024 was excluded due to satellite orbital adjustments.”

21. Page 25, line 481. Please indicate here at which rainfall rate (or rates) was set the precipitation detection threshold.

Response: We have updated the sentence to explicitly incorporate the precipitation detection threshold. The revised text now reads (page 25, lines 453-454):

“For precipitation detection (a binary classification task with a 0.1 mm/h threshold), ……”

22. Page 30, line 598. Fig. 9 not found.

Response: We sincerely appreciate the reviewer’s careful attention to figure references. Following the manuscript revisions, the original Fig. 8 has been renumbered as Fig. 10 due to the insertion of new figures in the preceding sections. The updated text now explicitly refers to Fig. 9 (previously Fig. 8).

23. Page 32, line 637. Is the ‘monitoring evaluation period’ the ‘validation’ period? Please use the same terms to avoid possible confusions.

Response: We appreciate the reviewer’s attention to terminology clarity. To avoid confusion, we have standardized the term “monitoring evaluation period” to “validation period” throughout the manuscript. Additionally, as part of our efforts to optimize the manuscript structure for Nature Communications format, we have moved the original “Systematic Evaluation of RePPIC-Net in Nowcasting” section into the “Performance of RePPIC-Net in Nowcasting” section. Consequently, the original phrasing has been removed from the main text.

24. Page 32, line 650 (and 652, 654-655). Please check English: were downloaded and downloadable?

Response: We thank the reviewer for highlighting the linguistic inconsistency in the original sentence. We have revised the sentence (page 35, lines 656-657):

“The GPM datasets were downloaded from the Goddard Earth Sciences Data and Information Services Center (GES DISC).”

25. Page 36, Fig. 6. Please add in the figure caption the IMERG version (Late?) as in Fig. 8.

Response: We thank the reviewer for suggesting the inclusion of the IMERG version in the figure caption. Following this advice, we have updated the caption of Fig. 7 to specify “IMERG-Late” (as in Fig. 8), ensuring consistency across figures.

26. Page 37, Fig. 7 caption. The content of panels a) and b) should be indicated.

Response: We appreciate the reviewer's feedback on clarifying the content of panels (a) and (b). The caption for Fig. 7 has been revised to explicitly indicate the content of each panel: "(a) Topographic map of the Tibetan Plateau and distribution of observation stations; (b) A snowfall event over the Tibetan Plateau at 19:00 UTC on 14 December 2023.".

Reviewer #2:

The motivation of the work is highly significant operational meteorology, hydrology, and disaster management because Real-time, accurate precipitation phase distinction is a long-standing global challenge, especially in data-sparse mountainous and high-latitude regions where hazards like avalanches and blizzards are common.

The paper proposes an interesting combination of AI architectures, fusing physics-informed meteorological fields from a rapid AI weather model (FuXi) with high-resolution satellite observations and a nowcasting model.

The main results in the paper are that the authors want to develop a system for real-time operations that achieves comparable rain-detection performance between their model and IMERG-Late with CSI scores of 0.95 and 0.96 and similar bias in snowfall overestimation and rainfall underestimation. In a given case study, the authors demonstrate the superior skill of the model in predicting a high impact snow blizzard, outperforming IMERG and FY-4B QPE product.

Major Critical Comments:

(1) The proposed training dataset seems quite small, training over June 2022 to September 2023, which covers likely only one snowfall season and one rainfall season, so it's unclear if there was any overfitting of the model.

Response: We sincerely thank you for raising this important point regarding the dataset. The primary reason for the relatively short training period (June 2022 to September 2023) is that the FY4B and its ground application system officially **began providing observation data and application services from June 1, 2022**. To ensure model robustness given this data constraint, we not only **extended the independent testing period until March 2025** to more thoroughly evaluate the model's generalizability, but also, we incorporated three key techniques in the model design to **effectively prevent overfitting**: 1) **Applied a sliding window technique** during sample construction to expand the dataset to 37,887 samples, significantly enhancing data diversity; 2) Carefully **controlled training epochs** (20 for the precipitation detection model, 100 for the precipitation retrieval model) to avoid overtraining; 3) **Opted for a simplified architecture** with three layers based on data characteristics, prioritizing generalization capability over complex structures. These measures collectively improved the model's robustness.

(2) When looking at the evaluation time period of October 2023 to January 2024, that again seems very small to draw any concrete conclusions from. It covers only the snowfall period and even then extending the evaluation period to include up until

March 2024 would be better.

Response: We sincerely thank you for these critical comments. We fully agree that an evaluation period of sufficient length is essential for verifying the model's generalizability. Accordingly, we have extended the independent evaluation period exclusively in the winter months (October–March of 2023–2025) when snow occurs, as phase discrimination is not required in rain-only periods.

(3) The evaluation should also cover the rainfall season so that we can understand any False alarm ratios better in cases when there is no rainfall as the model could be overfitting.

Response: We sincerely thank you for the suggestion. We would like to note that since this study **focuses specifically on the discrimination of precipitation phase (rain/snow)**, we concentrated the evaluation from October to March across 2023–2025 when precipitation phase is the most relevant and challenging. Seasons dominated by rainfall were therefore not included in this evaluation. In our model, temperature and humidity are most important factors for phase discrimination, and we have included the **stability performance of the FuXi fields** (e.g., temperature correlation >0.99, relative humidity correlation >0.96, new added Table 4) as supporting evidence of the robustness of the input data used in our model.

Meanwhile, to **better understand false alarm ratios**, we **added a new section "The spatiotemporal error distributions"**. We have systematically **evaluated the false alarm ratios across different stations**, further revealing the spatial distribution characteristics of the model's errors. Additionally, the newly added temporal analysis demonstrates stable performance across different months **similar to GPM IMERG**. It indicates that the model maintains stable performance on this multi-year, independent dataset, demonstrating its robustness, rather than overfitting to the limited training set.

Table 4 Errors in the 12-hour FuXi forecasts and ERA5 reanalysis for meteorological variables used in precipitation retrieval models.

Variable	Bias	RMSE	CC
T500	-0.02	0.44	1.00
T700	0.02	0.57	1.00
T850	0.10	0.90	1.00

T925	0.09	1.01	1.00
R500	-0.35	8.49	0.96
R700	-0.58	7.58	0.96
R850	-0.81	6.79	0.96
R925	-0.61	6.80	0.96
U500	-0.04	1.42	0.99
U700	0.03	1.26	0.98
U850	0.03	1.07	0.98
U925	0.03	0.84	0.98
V500	0.04	1.40	0.99
V700	-0.08	1.26	0.98
V850	0.05	1.09	0.98
V925	0.06	0.94	0.98
TP	0.28	1.13	0.80

(4) The paper claims to be highly performant for extreme precipitation events but only shows CSI for values up to 5mm/hr and only showcases one case study for a high impact event. From this case study it's really hard to draw any conclusions that the system is indeed highly performant over different scenarios.

Response: We thank the reviewer for this critical comment. We agree that the use of "extreme precipitation" was incorporate. In response, we have **revised ".....notably for extreme precipitation events" to ".....notably for quantifying mixed-phase precipitation intensity". And revise the subsection title from "Performance of RePPIC-Net in Extreme Events" to "Case studies of mixed-phase and orographic snowfall" to more accurately reflect the content and focus.**

We would like to clarify the primary focus of our study: the accurate identification of precipitation phase (rain/snow), particularly near the critical transition point. Such events often have **lower intensity but are critically important for disaster warning** (e.g., snowstorms). Thus, the key innovation lies in addressing the monitoring challenge for complex-phase, low-intensity yet high-impact solid precipitation that traditional methods often miss, rather than emphasizing capabilities for extreme rainfall events.

We fully agree that a single case study is insufficient for drawing general conclusions. The Tibetan Plateau case was also included in the case studies and we added a new section titled “The spatiotemporal error distributions” in the revised manuscript. The Tibetan Plateau case tests a completely different topography, while the spatiotemporal error analysis shows **performance is location-dependent**. Most importantly, the **case study’s CSI aligns with the regional average**, proving its representativeness. This collectively provides stronger evidence for consistent performance across varying conditions.

The new systemic analysis (The spatiotemporal error distributions) also reveals the model's performance stability and limitations from both spatial and temporal dimensions: 1). **Spatially**, the model shows superior snow discrimination performance in northern China compared to southern regions, highlighting the **persistent challenge of phase discrimination under near-freezing conditions in the south (Fig. 3)**. 2). **Temporally**, our model shows a **similar monthly CSI variation trend to GPM IMERG (Fig.4)**. These systematic evaluations strengthen our understanding of the model's capabilities, limitations, and **future improvement directions**.

Fig. 3 Spatial distribution maps of CSI, POD, and FAR for RePPIC-Net and GPM IMERG-Late.

Fig. 4 Temporal variations of CSI and Frequency Bias for RePPIC-Net and GPM IMERG-Late. Performance of RePPIC-Net in extreme events.

(5) Clarification of the "Real-Time" Capability: The central claim of the paper is its "real-time" operational capability, which is highlighted as a key advantage over systems with a "4-hour latency". However, the methods section states that the FuXi model is "initialized with ERA5 data (about 8-hour latency)". This creates a significant contradiction. While the FuXi model's forecast generation is computationally fast ("within seconds"), its dependency on an 8-hour-old initial state seems to undermine the "real-time monitoring" claim. The authors must clarify this point. Is the system truly real-time, or is it more accurately described as a rapid-update system with an 8-hour data lag? A detailed discussion of how this latency in the input data affects the final product's timeliness and accuracy is essential for the paper's credibility.

Response: We sincerely thank the reviewer for raising this critical point regarding the real-time capability of our framework, which allows us to clarify a potentially misleading aspect of our methodology description.

The reviewer is correct in noting that the FuXi model was trained and validated with ERA5 reanalysis data. However, in actual deployment, our system can be **initialed by the ECMWF Operational products (OPs)**, which is a near-real-time product with a latency of approximately 8 hours. We therefore configured the FuXi model to produce a **12-hour forecast**. This setup is pivotal for our operational application, as it allows the system to generate forecasts of the critical **0-3 hours nowcasting window**. This design effectively bridges the latency gap. Therefore, when a user requests a forecast, the system utilizes the most recent OPs (8 hours old) and generates a forecast for the next

0-3 hours within **approximately 2 minutes of computation**. This workflow genuinely enables real-time monitoring and short-term forecasting.

We have revised the manuscript to include a detailed clarification in the **Implementation Details (pages 28-29, lines 508-543), where we not only quantify the computational runtime of our model but also provide a detailed explanation of its operational capability for real-time application.**

“Implement details

All model training and inference were conducted on the Computing for the Future at Fudan (CFFF) high-performance computing platform. Each computing node was equipped with two AMD EPYC 7H12 64-core processors and one NVIDIA RTX A6000 GPU (50 GB memory). The software environment consisted of Ubuntu 22.04 LTS, Python 3.10, PyTorch 2.0.0, and CUDA 11.8. The models were trained using AdamW optimizers, with learning rates of 1×10^{-4} for UNet/ResUNet and 1×10^{-5} for the DAYU-FY model. Batch sizes were set to 16 (for UNet/ResUNet) and 1 (for DAYU-FY). All experiments were performed under identical environments to ensure reproducibility.

Table 1 summarizes the computational complexity (FLOPs) and parameter counts (Parameters) of the major sub-models in the RePPIC-Net framework. The UNet and ResUNet modules are responsible for precipitation phase classification and quantitative precipitation retrieval, respectively. The FuXi short model generates real-time atmospheric physical fields, while the DAYU-FY model performs satellite brightness temperature (BT) extrapolation for precipitation nowcasting. Measured on an NVIDIA A6000 (50 GB) GPU setup, a single end-to-end full-disk (0.05°) inference combining precipitation phase classification and precipitation retrieval takes approximately 77.41s; The subsequent PLP calculation for phase discrimination requires an additional 2s. The runtime of the upstream forecasting models that supply inputs is: the FuXi short model takes about 5.1s to produce a 6-hour forecast, and the DAYU-FY model takes about 2s to produce a 30-minute BT extrapolation. All timings are measured under the hardware/software environment stated in Methods and represent typical inference times for the reported resolution and product. Thus, a real-time forecast can be completed within 2 minutes.

Furthermore, we have provided a detailed explanation on the real-time operational capability of our product. The GPM IMERG Early product, which requires the integration of multi-orbit and multi-sensor satellite observations and relies on meteorological fields from conventional numerical models, exhibits a latency of approximately 4 hours. In contrast, although our model was trained and validated with ERA5 reanalysis data with one-week delay, it can operationally utilize near-real-time ECMWF Operational

Products (Ops), which have a latency of about 8 hours. By incorporating the FuXi 12-hour forecast, meteorological fields covering approximately up to 4 hours forecasts are obtained. This configuration enables our system to generate real-time precipitation monitoring and 1-3 hour forecast products in approximately 2 minutes of computation."

(6) The paper claims accuracy at 0.05 degree spatial resolution but all results are against IMERG data which is at 0.01 degree resolution, so it's unclear what the validation mechanism is for accuracy at 0.05 degrees.

Response: Thank you for this important question regarding spatial resolution and validation. We would like to clarify that our product's 0.05° resolution is fundamentally derived from the training target, the **GPM DPR product (0.05°)**. Most importantly, the validation of our product's accuracy at this 0.05° resolution was performed against a completely independent dataset: **more than 2000 ground stations across China**.

Primary Validation (for 0.05° resolution claims): Our claims of accuracy at 0.05° resolution are derived from a direct comparison against **ground station observations**. Each station has precise longitude and latitude coordinates. We compare the predicted value from the specific 0.05° grid cell in which station is located (without any interpolation). Therefore, this core accuracy assessment is independent of the IMERG data and its native resolution.

Comparison with IMERG (Appendix): Given the lack of ground station data beyond China, the full-disk comparison with IMERG (0.1°) in the appendix is intended for qualitative, large-scale pattern analysis. To enable a fair comparison on the same grid, we remapped the IMERG data to our 0.05° grid using bilinear interpolation.

(7) The authors introduce a novel and useful composite metric, the Critical Success Index for Phase and Detection (CSI-PD), defined as $CSIP \cdot CSID_n$. While the rationale is explained as penalizing models that are good at one task but not the other, the specific choice of multiplication versus other methods of combination (e.g., a weighted average) is not justified. A brief sentence explaining why this multiplicative, probabilistic formulation was chosen would strengthen the methodological contribution.

Response: Thank you for the valuable feedback. We adopted the multiplicative operation based on a probability theory foundation: CSI-PD aims to **quantify the joint probability of a model simultaneously correctly judging phase and detecting precipitation**, i.e., $P(\text{Phase} \cap \text{Detection})$. The probabilistic basis of this metric can be expressed as $P(\text{Phase}) \cdot P(\text{Detection})$ under the **independence assumption**. The multiplicative structure offers a key advantage: it **requires the model to perform well on both tasks simultaneously**. Failure in either sub-task (e.g., a detection failure

resulting in $CSI_D^n = 0$) will cause the composite metric to fail entirely ($CSI-PD = 0$). This aligns with the practical requirement in blizzard event monitoring where "**a failure in detecting the occurrence of precipitation or a failure in phase identification would both result in a failure to monitoring a blizzard.**" Weighted averaging or other linear methods cannot capture this critical constraint. Therefore, the multiplicative approach is more consistent with the design intent of CSI-PD. Additionally, we have further refined the explanatory section in the original text accordingly.

We have revised the original sentence from "The joint optimization ensures balanced performance in both phase classification and detection tasks" to "The multiplicative combination enforces joint optimization, ensuring balanced performance across both tasks."(page 9, lines 166-167).

Reviewer #3 (Remarks on code availability):

The code appears to be well-written and free of technical issues.

Reviewer #3 (Remarks to the Author):

Summary:

The manuscript addresses the real-time monitoring of precipitation phase and intensity by integrating AI-driven atmospheric forecasts (FuXi), geostationary satellite observations (FY-4B), and auxiliary geolocation information. The authors employ a U-Net architecture for phase discrimination and a ResU-Net for intensity estimation, trained using GPM DPR products as reference labels. The topic is timely and relevant for advancing real-time precipitation monitoring, especially over data-sparse regions such as the Tibetan Plateau.

However, the manuscript has **major weaknesses in terms of novelty, methodological rigor, and the positioning for a high-impact science journal of *Nature Communications***. Below are the specific points of evaluation.

Strengths:

- 1. Timely Topic:** Real-time phase-intensity discrimination is highly relevant for both weather and climate communities. Performance evaluation over challenging environments such as the Tibetan Plateau adds practical value and scientific interest.
- 2. Integration of Multiple Data Sources:** Combining AI-driven atmospheric forecasts (FuXi), geostationary observations (FY-4B), and probabilistic classification methodology shows an attempt toward an integrated solution.
- 3. Operational Relevance:** Emphasis on low-latency capability (leveraging FuXi forecasts and FY-4B observations), and short-term precipitation nowcasting (1–3 hours lead time) enhances the practical utility of the system.

Major Concerns:

1. Novelty and Conceptual Contribution

The core methodological contributions appear incremental rather than transformative. The use of Unet and ResUNet architectures for precipitation phase and intensity estimation is well-established in remote sensing and computer vision domains, and binary cross-entropy loss is a standard approach for classification problems. Furthermore, the phase classification algorithm relies heavily on the existing GPM-IMERG PLP framework, raising concerns about whether the approach represents a novel methodology or a repurposing of an established algorithm. The study risks being

perceived as an application note rather than a fundamental advance.

Response: We sincerely thank the reviewer for this insightful comment. We agree that the individual technical components, such as U-Net/ResUNet and cross-entropy loss, are well-established. Our contribution is to propose a novel hybrid AI model addressing a critical gap in surface precipitation phase monitoring and nowcasting. We show that **the integration of advanced AI-weather model and AI-precipitation model offers a timely solution for this long-standing issue, and this hybrid framework is fundamentally novel**. This paper presents a blueprint for optimizing AI models in atmospheric science, which are growing successively in number and complexity, amid a scarcity of coordinated applications.

Technically, the AI-precipitation module is developed specifically for this hybrid model, with the following contributions:

1). Precipitation Model Optimization: To address the inherent limitations of IR-based precipitation retrieval—particularly the lack of vertical profiling information—we introduced three key innovations, including a **rain cluster spatial feature extraction module, a probabilistic post-processing module, and the integration of vertical meteorological fields**. Collectively, these designs enable our precipitation retrieval model to outperform not only traditional IR-based methods (such as the method used in the operational FY-4B QPE) but also widely adopted machine learning techniques like random forest. The comparisons have been added into the text (page 26, lines 472-478) as below:

“Our deep learning framework with meteorological field inputs achieved a CSI of 0.38 (at 0.1 mm/h threshold), significantly outperforming both point-to-point random forest approaches (0.31) and satellite-only deep learning configurations (0.33). Notably, even the random forest method augmented with meteorological fields reached only 0.36, underscoring the distinct advantage of our integrated architecture that combines area-based processing with atmospheric profile information.”

2). Pioneering Nowcasting Model: We developed the first short-term IR brightness temperature extrapolation model (DaYu-FY) trained on FY-4B AGRI data. Its hybrid ResUNet and Pyramid Vision Transformer (PVT) architecture is designed to precisely balance local details with long-range spatiotemporal dependencies.

As for the reviewer’s concern on the GPM-IMERG PLP framework, we acknowledged that **T_w is widely recognized as the most robust thermodynamic variable for phase discrimination** (e.g., Wilson et al., 1941; Prein et al., 2020). And phase discrimination

based on surface meteorology has its limitations even under ideal conditions (Jennings et al., 2025). Our value lies in providing an optimized solution that is operational in real-time and nowcasting. Unlike GPM PLP framework, which uses surface T_w , we **compute T_w from FuXi's forecasting atmospheric profiles** at the pressure level matching the local terrain height (due to the lack of surface humidity data). Also, a core difference, enabled by FuXi's rapid forecasting capability, **allows us to overcome GPM PLP's limitations (at least 4h latency), offering real-time operation and forecasting capability.**

New text to clarify our precipitation phase discrimination method (pages 21-22, lines 376-381):

"Central to this module is the estimation of wet-bulb temperature (T_w), a thermodynamic variable recognized in previous studies as the most robust indicator for precipitation phase discrimination³⁹⁻⁴⁰. Its established superiority over traditional approaches—including those based on dew point temperature, surface air temperature (T_a), and even certain machine learning techniques⁴¹."

Wilson, W. T. An outline of the thermodynamics of snow-melt. *Eos Trans. Am. Geophys. Union* 22,182-195 (1941).

Prein, A.F., Heymsfield, A.J. Increased melting level height impacts surface precipitation phase and intensity. *Nat. Clim. Chang.* 10, 771–776 (2020).

Jennings, K. S., et al. Machine learning shows a limit to rain-snow partitioning accuracy when using near-surface meteorology. *Nat. Commun.*16(1), 2929 (2025).

2. Dependence on Latency and Real-Time Feasibility

The proposed system claims to enable real-time phase discrimination by combining FuXi forecasts (with ERA5 initialization) and geostationary satellite data. However, ERA5 initialization introduces an approximate 8-hour latency, which contradicts the claim of real-time operation. This undermines the practicality and uniqueness of the proposed framework compared to operational systems.

Additionally, the reliance on FuXi's 12-hour forecasts and the inherent uncertainties in short-term atmospheric model outputs could introduce systematic biases, although there are optimization corrections, it has not fully addressed in the manuscript. The authors should rigorously quantify how forecast errors propagate into phase discrimination and intensity estimation, particularly under extreme or rapidly evolving weather events.

Response: We thank the reviewer for this insightful observation. We have clarified the distinction between the model's training and operational periods to resolve the apparent contradiction regarding real-time ability. During **training and validation period**: To build a high-quality training dataset, we used ERA5 reanalysis data to drive the FuXi model, generating historically consistent atmospheric states aligned with past satellite observations. This stage does not require real-time data, as its goal is to learn a robust mapping from "atmospheric thermodynamics + satellite observations → precipitation." During **operational period**: In real-time operation, RePPIC-Net **can be initialized by real-time operational global analysis fields (e.g., ECMWF Operational analysis products)**, which have a latency of typically < 8 hours. These fields are specifically designed for operational numerical weather prediction. When combined with the FuXi 12-hour forecasting ability, this setup enables the generation of precipitation products covering approximately up to +4 hours relative to analysis time. We have added a new subsection in the Appendix (**Implementation Details**) to explicitly describe the operational workflow and latency budget, ensuring readers will not be misled.

Additionally, we have **expanded our analysis to quantify how FuXi forecast uncertainties (Table 4)** propagate into phase discrimination and intensity estimation in **three ways**:

1). Quantification of Error Propagation: We designed a **comparative experiment where RePPIC-Net was driven separately by ERA5 reanalysis data (representing the "true" atmospheric state) and FuXi's 12-hour forecasting data**. This allowed us to systematically quantify how initial condition and forecast errors from FuXi propagate and impact the final accuracy of phase discrimination and precipitation intensity estimation.

2). Evaluation for Rapidly Evolving Events: We are confident that **the extended long-term evaluation period (spanning two full winters, totally 11 months)** inherently encompasses various rapidly evolving weather events. We specifically analyzed the model's performance within 0-5mm/h precipitation to better focus on the mixed-phase precipitation.

3). Insight into Physical Mechanisms: We found that quantifying the impact of different meteorological inputs on the precipitation results also **provides valuable insights into RePPIC-Net's internal decision-making process**. For instance, the model's sensitivity to key thermo-dynamic parameters (e.g., temperature) aligns with our physical understanding of precipitation. This analysis enhances the model's physical interpretability.

We have **added a quantitative uncertainty assessment of the FuXi forecasts, along with a related physical interpretation, as follows (pages 32-33, lines 611-633):**

“Table 4 presents the error analysis in the 12-hour FuXi forecasts and ERA5 reanalysis for meteorological variables used in precipitation retrieval models. FuXi meteorological fields demonstrate high overall accuracy, with temperature estimates exhibiting a higher correlation coefficient than relative humidity. With the national automatic weather station network in China, we evaluated the discrepancies of FuXi and ERA5 from October to March, 2023-2025. We found a systematic overestimation over temperature (except at 500 hPa) and underestimation of relative humidity in FuXi (across all pressure levels).

The comparable performance of FuXi and ERA5 results in the similar precipitation phase and detection (Table 5). However, a slight systematic bias observed by FuXi leads to simultaneously lower Probability of Detection (POD) and False Alarm Rate (FAR) of precipitation detection for snow and rain. This pattern reveals that the precipitation model has an inherent tendency to predict more precipitation events under the specific thermodynamic profile of ERA5. This can be attributed to ERA5's generally lower temperatures (except at 500 hPa) and higher humidity at all levels—conditions that favor precipitation generation, particularly given the high feature importance of relative humidity at 700 hPa (R700) in our model. Furthermore, the ERA5-driven model exhibits a greater propensity to predict snow. This is likely due to its lower near-surface temperatures, which yield lower wet-bulb temperatures. The FuXi's slight warm bias appears to partially compensate for the inherent cool bias introduced by using pressure levels as a proxy for the actual surface conditions. Notably, inherent uncertainties in ERA5 within cloud systems may contribute to these differences⁴⁸.”

Table 4 Errors in the 12-hour FuXi forecasts and ERA5 reanalysis for meteorological variables used in precipitation retrieval models.

Variable	Bias	RMSE	CC
T500	-0.02	0.44	1.00
T700	0.02	0.57	1.00
T850	0.10	0.90	1.00
T925	0.09	1.01	1.00

R500	-0.35	8.49	0.96
R700	-0.58	7.58	0.96
R850	-0.81	6.79	0.96
R925	-0.61	6.80	0.96
U500	-0.04	1.42	0.99
U700	0.03	1.26	0.98
U850	0.03	1.07	0.98
U925	0.03	0.84	0.98
V500	0.04	1.40	0.99
V700	-0.08	1.26	0.98
V850	0.05	1.09	0.98
V925	0.06	0.94	0.98
TP	0.28	1.13	0.80

Table 5 Evaluation of precipitation phase and detection using different meteorological field inputs.

		POD	FAR
Phase	ERA5	0.88	0.06
(rain)	FuXi (12h forecast)	0.91	0.08
Phase	ERA5	0.87	0.25

(snow)	FuXi (12h forecast)	0.83	0.19
Detection	ERA5	0.54	0.53
(rain)	FuXi (12h forecast)	0.53	0.45
Detection	ERA5	0.29	0.53
(snow)	FuXi (12h forecast)	0.27	0.41

3. Validation Strategy and Robustness of Results

The study uses GPM DPR as the reference for both precipitation phase and intensity, but this raises several concerns. First, GPM DPR's revisit time and sampling limitations make it an imperfect ground truth for evaluating full-disk geostationary-based predictions. Second, the performance evaluation primarily focuses on statistical accuracy over large areas and special regions (e.g., the Tibetan Plateau), but there is limited analysis of failure cases or uncertainty quantification in heterogeneous terrains and convective systems. Furthermore, the validation appears superficial and lacks comprehensive metrics, and temporal consistency checks. Without robust uncertainty analysis, the reliability of the integrated system for operational purposes remains unclear.

Response: We sincerely thank the reviewer for these constructive comments. We have strengthened our analysis and discussion in response to your points.

1). On the Suitability and Limitations of GPM DPR as a Reference Standard

The reviewer correctly highlights the inherent limitations of GPM DPR—namely its revisit time and sampling characteristics—as a reference for validating full-disk, high-frequency geostationary-based products. **We fully acknowledge the core challenge that no "perfect ground truth" exists.** DPR was adopted for training in this study because, as an active spaceborne radar, it is widely regarded as **the most reliable available benchmark for providing global precipitation in this context.** Also, we have incorporated **ground station observational data in our primary validation** and **GPM IMERG for full-disk evaluation** to establish a more robust evaluation framework which accounts for the limitations in GPM DPR's revisit time and sampling.

2). On In-Depth Analysis of Complex Scenarios and Failure Cases

We agree that a deep understanding of a model's failure cases is as important as highlighting its successes. The revised manuscript enhances the analysis of model performance over heterogeneous terrain and within complex weather systems through two ways:

First, we add the Tibetan Plateau case in the “Case Studies of Mixed-Phase and Orographic Snowfall” section, which indicates that both GPM IMERG and our model face detection challenges in this region, highlighting the **limitations** of current satellite-based precipitation over such complex underlying surfaces;

Second, our newly added systematic analysis of **error spatial distribution reveals that the model exhibits weaker performance in discriminating snow phase under near-freezing rain conditions in southern China**. These systematic error patterns are not random failures but are closely linked to specific geographic and meteorological factors, providing clear direction for future improvements. For example, in Tibetan Plateau, while phase discrimination capability remains robust (CSI > 0.9), significant challenges persist in overall precipitation detection, as illustrated in Fig. 10 (CSI < 0.1).

Fig. 10 Spatial distribution maps of CSI, POD, and FAR of precipitation detection for RePPIC-Net and GPM IMERG-Late.

3). On Comprehensiveness of Validation: Integrated Metrics, Temporal Consistency, and Uncertainty Quantification

We have improved the validation section in following three ways: **First, add comprehensive metrics**. Beyond CSI, we have systematically improved the evaluation with a suite of metrics including False Alarm Ratio (FAR), Probability of Detection (POD), and Frequency Bias (FB) to assess model performance across multiple dimensions; **Second, we have now performed uncertainty assessment on key verification scores**

(Fig. 2b); **Third**, we adopted the reviewer's suggestion for a "temporal consistency check." In the new "Spatiotemporal Error Distribution Characteristics" section, analysis of the model's performance across consecutive precipitation events demonstrates that **RePPIC-Net and GPM IMERG share highly consistent monthly variation trends (Fig. 4)**. This indicates the temporal robustness of our model, comparable to GPM IMERG-Late, indirectly supporting its reliability for potential operational applications.

We believe these additions significantly strengthen the robustness and credibility of our conclusions and provide a more profound and honest assessment of the model's capabilities and limitations.

(b)

		GPM IMERG		RePPIC-Net	
		CSI	Uncertainty(\pm)	CSI	Uncertainty(\pm)
Phase	Rain	0.8252	0.0004	0.8477	0.0004
(CSI _P)	Snow	0.7099	0.0005	0.6896	0.0007
Detection	Rain	0.3713	0.0017	0.3713	0.0018
(CSI _D)	Snow	0.1411	0.0034	0.2283	0.0041
Overall	Rain	0.3064	0.0014	0.3147	0.0015
(CSI-PD)	Snow	0.1001	0.0024	0.1574	0.0028

Fig. 2| (b) CSI_P, CSI_D, CSI-PD and uncertainty of RePPIC-Net and GPM IMERG within the 0.1–5 mm/h precipitation intensity range.

Fig. 4 Temporal variations of CSI and Frequency Bias for RePPIC-Net and GPM IMERG-Late. Performance of RePPIC-Net in extreme events.

4. Positioning Against Existing Systems

The manuscript compares the proposed system to FY-4B and GPM IMERG-Late products, but these comparisons are primarily descriptive and do not convincingly demonstrate a clear performance advantage in terms of accuracy, latency, or operational applicability. The added value of integrating FuXi AI-driven forecasts with satellite observations needs to be explicitly quantified against state-of-the-art precipitation phase/intensity retrieval algorithms. In its current form, the manuscript does not sufficiently justify why the approach is superior to existing multi-source fusion techniques, limiting its perceived novelty and impact.

Response: We thank the reviewer for this critical comment. We have undertaken major revisions to the manuscript to convincingly.

1). Clarifying the Performance Advantage: Accuracy, Latency, and Operational Applicability

To systematically demonstrate the performance advantages, we have implemented the following improvements in the revised manuscript:

Accuracy: The analysis in the new section "The spatiotemporal error distributions" demonstrates that RePPIC-Net achieves overall precipitation discrimination accuracy comparable to GPM IMERG-Late and is slightly superior for snowfall, with CSI-PD scores of 0.1574 versus 0.1001 (Fig. 2b). FY-4B QPE showed the lowest accuracy in the

full-disk evaluation.

Latency and Operational Applicability: We have explicitly quantified the system's latency in the "**Implementation Details**" section: RePPIC-Net generates products within approximately 2 minutes, and it's currently the **only system capable of providing real-time nowcasting** (1-3 hours) with phase discrimination. In contrast, the GPM IMERG product has a latency of at least 4 hours with FY-4B QPE of about 30 minutes. While FY-4B QPE offers good timeliness, it provides **no phase products, so** we have performed the comparison with FY-4B QPE in the appendix, presenting it as a **baseline** for full-disk performance evaluation.

2). Explicitly Quantifying the Added Value of Integrating FuXi Forecasts

We have directly quantified the contribution of integrating the FuXi meteorological forecasts through experiments and attribution analysis:

Precipitation phase: FuXi demonstrates superior performance over traditional numerical models in synoptic-scale circulation forecasting, achieving real-time computational speed within seconds (Chen, L. et al. 2023). Also, FuXi provides the essential temperature and humidity fields required for phase discrimination, which is the fundamental basis for achieving **real-time precipitation phase monitoring and nowcasting**.

Precipitation intensity: Comparative experiments show that introducing the FuXi meteorological fields significantly improves the CSI for precipitation detection from 0.36 to 0.38. The corresponding description has been included in the text (page 26, lines 472-478):

"Our deep learning framework with meteorological field inputs achieved a CSI of 0.38 (at 0.1 mm/h threshold), significantly outperforming both point-to-point random forest approaches (0.31) and satellite-only deep learning configurations (0.33). Notably, even the random forest method augmented with meteorological fields reached only 0.36, underscoring the distinct advantage of our integrated architecture that combines area-based processing with atmospheric profile information."

Validation of Physical Mechanism: Feature importance analysis (see Table 2) indicates that variables provided by FuXi, such as Total Precipitation (TP) and Relative Humidity at 500 hPa (R500), rank among the most contributory in the model. This physically validates their value, showing that the model's learning mechanism aligns with atmospheric thermodynamic principles.

Chen, L. et al. FuXi: A cascade machine learning forecasting system for 15-day global weather forecast. npj Clim. Atmos. Sci. 6, 190 (2023).

Table 2 Importance scores of variables of Precipitation Detection and their corresponding rankings.

Variables Score	Ranking	Variables Score	Ranking
$\Delta T_{6.25-10.8} = 0.072$	1	$BT_{12.0} = 0.0352$	15
$\Delta T_{7.42-12} = 0.072$	2	R850 = 0.0328	16
TP = 0.058	3	$BT_{10.8} = 0.0328$	17
R700 = 0.048	4	Elevation = 0.0319	18
$BT_{8.55} = 0.046$	5	R925 = 0.0317	19
R500 = 0.045	6	$BT_{13.3} = 0.0313$	20
$\Delta T_{10.8-12} = 0.044$	7	Div500 = 0.0261	21
Longitude = 0.040	8	DIV850 = 0.0254	22
T500 = 0.0373	9	DIV700 = 0.0253	23
T700 = 0.0366	10	DIV925 = 0.0251	24
T925 = 0.0365	11	$BT_{7.42} = 0.0245$	25
T850 = 0.0361	12	$BT_{6.25} = 0.0222$	26
SAZ = 0.0358	13	$BT_{6.95} = 0.0218$	27
Latitude = 0.0352	14		

5. Scientific Insight vs. Algorithmic Engineering

While the technical implementation is sound, the study reads more like an engineering solution for operational nowcasting rather than a contribution that advances scientific understanding of precipitation processes. The manuscript lacks physical interpretability and mechanistic insight. Does FuXi better capture the 0°C isotherm or wet-bulb temperature structure? How does this improvement manifest across different synoptic regimes, e.g., warm vs. cold precipitation systems, freezing rain events, orographic snowfall? The authors could strengthen its scientific contribution by analyzing how integrated atmospheric forecasts and satellite signals improve phase discrimination under challenging physical conditions (e.g., mixed-phase precipitation, topographic influence on snow/rain transition), or by proposing a novel interpretable AI architecture informed by atmospheric physics. Currently, the discussion of underlying physical

processes and their interaction with the AI-driven framework is limited.

Response: We sincerely thank the reviewer for this insightful and constructive suggestion. **Guided by your comments**, we have revised the manuscript to enhance its scientific contribution. **Our enhanced analysis provides deeper insights into how the integrated AI framework captures and quantifies the key physical processes governing precipitation.**

Key enhancements include:

1). Systematic Validation of Physical Consistency

The new section "relationship between meteorology and precipitation phase" **improve model interpretability via feature importance ranking and quantify the impacts of meteorology on precipitation.** Crucially, it reveals that our model's decision-making is highly consistent with established atmospheric physics. For instance, through controlled experiments, we demonstrate that colder and more humid atmospheric conditions systematically increase the model's probability of predicting precipitation occurrence and its phase as snow.

2). FuXi Performance in Capturing Critical Thermodynamic Structures

Our analysis confirms that the FuXi forecasting **maintain high consistency** with ERA5 reanalysis in thermodynamic variables (temperature correlation > 0.99; relative humidity correlation > 0.96). FuXi and the Japan Meteorological Agency (JMA) forecast-based data used in GPM IMERG-Late demonstrate **comparable skill** in representing the overall wet-bulb temperature structure: it achieves a higher CSI for rain but a slightly lower CSI for snow phase discrimination. Notably, FuXi shows a frequency bias closer to 1, indicating a more balanced representation of wet-bulb temperature.

3). The improvement manifest across different synoptic regimes

While our current work primarily focuses on establishing an integrated framework for precipitation grounded in robust physical consistency and interpretation, we recognize the value of analyzing performance variations across distinct synoptic systems. To address this, we conducted **spatial error analysis** (Fig. 3. and Fig. 10) to isolate model performance under different precipitation regimes: regarding precipitation phase, **cold precipitation dominated regions** (e.g., northern China) shows better performance than warm precipitation dominated regions (Fig. 3c); Regarding precipitation detection, warm precipitation dominated regions (e.g., southern China) shows better performance than cold precipitation dominated regions (Fig. 10c); In **orographic precipitation** (e.g., Tibetan Plateau), phase discrimination capability remains relatively strong (Fig. 3c), but precipitation detection performance still faces challenges (CSI < 0.1, Fig. 10c). In

freezing rain events, given their low occurrences and the challenge of modeling such complicated thermodynamics, we will conduct further research in subsequent studies. We add in the conclusion (page 20, lines 352-356):

“Future efforts will focus on augmenting microwave satellite constellations to improve precipitation intensity accuracy, and improve our model's phase discrimination capability under complex phase transitions (e.g., freezing rain, ice pellet).”

Fig. 3| Spatial distribution maps of CSI, POD, and FAR for precipitation phase from RePPIC-Net and GPM IMERG-Late.

Fig. 10| Spatial distribution maps of CSI, POD, and FAR for precipitation detection from RePPIC-Net and GPM IMERG-Late.

Added the new section as below (pages 31-32, lines 583-610):

“Relationship between meteorology and precipitation phase

To understand the physical mechanisms of the model and the influence of meteorological fields on precipitation phase, we first identified the most influential

meteorological variables for our precipitation model via a random forest feature importance analysis (Tables 2, 3). Following this identification, we analyzed the systematic errors in the 12-hour FuXi forecast fields relative to ERA5 reanalysis data over China (Table 4). Subsequently, we investigated the impact of perturbing these top-ranked variables on precipitation.

The relative importance of contributing features in precipitation detection was systematically evaluated through quantitative analysis (Table 2). Satellite channel differences emerged as highly significant predictors, as they effectively capture the essential changes in cloud properties and atmospheric conditions, which are fundamental for accurate precipitation detection. Among meteorological parameters, total precipitation shows the greatest impact on precipitation detection due to its direct reflection of the intensity and distribution of rainfall events. Additionally, the relative humidity at 700 hPa and 500 hPa ranked prominently, highlighting the critical role of mid- to upper-tropospheric water vapor advection in modulating precipitation initiation. The relative importance of contributing features in quantitative precipitation estimation was also systematically evaluated (Table 3). Notably, cloud cluster characteristics—specifically the distance to cloud boundary and the minimum 10.8 μm infrared brightness temperature within clusters—were found to exhibit high importance rankings, suggesting that incorporating cloud cluster information enhances quantitative precipitation estimate accuracy. Additionally, lower-level divergence and upper-level humidity parameters occupied prominent positions in the ranking hierarchy, indicating their critical roles in modulating precipitation intensity. Conversely, single-channel infrared satellite data consistently demonstrated lower predictive importance, highlighting the limited utility of isolated infrared spectral information for quantitative precipitation retrieval.”

Minor Comments:

1. The overall organization of the manuscript requires further improvement. The manuscript's structure is somewhat disorganized, and the presentation lacks clarity. Some content is repeated across sections, giving the text a redundant or “AI-generated” feel. In addition, some section of the manuscript is described in a way that is not sufficiently detailed or precise, making it challenging for readers to fully understand the methodology, assumptions, and experimental procedures.

Response: We thank you for your important feedback. To improve the manuscript's overall organization, content quality, and linguistic expression, the specific revisions are as follows:

1). Comprehensive Restructuring and Content Enhancement

We have **added systematic spatiotemporal error analysis, quantified the impact of ERA5 inputs, clarified the real-time operational mechanism, and enhanced physical interpretability analysis** to enhance the content.

2). Systematic Elimination of Redundancy

We conducted a section-by-section review of the entire text, merging or **removing similar descriptions** (e.g., repeated explanations of Tibetan Plateau case). This ensures that each point is presented only once within its most relevant context, making the content more concise and focused.

3). Enhanced Precision and Academic Rigor

We have rewritten all previously vague or insufficiently technical descriptions (e.g. the new added section "Implement Details") to ensure clarity. Furthermore, we specifically invited two senior colleagues within the field to perform multiple rounds of language polishing and academic review to effectively eliminates any "AI-generated" tone.

2. Using binary cross-entropy for precipitation intensity estimation is conceptually flawed, as precipitation intensity is inherently a continuous variable with a strongly right-skewed and zero-inflated distribution. Converting it into discrete classes for classification and applying binary or multi-class cross-entropy (BCE) results in a loss of important continuous information. This discretization introduces sensitivity to arbitrary thresholds and can lead to calibration biases, ultimately limiting the model's ability to accurately represent the true intensity distribution.

Response: We sincerely thank the reviewer for this insightful suggestion. In fact, during our practical experiments, **we have compared different loss functions, including Root Mean Square Error (RMSE)**. However, maybe due to the uncertainties in infrared-based precipitation retrieval, the extremely imbalanced distribution characteristics of precipitation intensity, and the U-Net architecture employed for image-to-image prediction, the regression model tended to converge to a "conservative" average state with loss function of RMSE. This made it difficult to accurately predict the precipitation intensity, particularly for moderate and heavy precipitation events. In contrast, although binary cross-entropy (BCE) loses some continuous information, it demonstrated advantages in this specific context: **larger precipitation magnitudes are physically consistent with higher precipitation occurrence probabilities**. This enables BCE to indirectly yet more robustly capture precipitation intensity.

We have revised the methodological description in the manuscript to clarify the reason

for choosing BCE (page 25, lines 455-460):

"For precipitation intensity retrieval, we employed a binary cross-entropy loss. This approach frames the retrieval of intensities exceeding a defined threshold as a probabilistic classification task, which demonstrated superior robustness to the inherent uncertainties in IR retrieval and the highly skewed distribution of precipitation values compared to regression losses (e.g., RMSE)."

Response Letter

Reviewer 1 (Remarks to the Author):

I thank authors for the corrected version and the detailed item-by-item response provided. I think the new corrected version fulfilled all items raised so it can now be accepted for publication.

Response: We sincerely thank you for your careful review and for confirming that our revised manuscript has adequately addressed all previous concerns. We are grateful for your time and insightful input throughout this process.

Reviewer 2 (Remarks to the Author):

Thanks for providing detailed responses to the reviewer comments and updating the manuscript accordingly. Based on the updates, the paper seems to have been significantly strengthened with robust results. The authors have effectively addressed the critical concerns regarding dataset size, operational latency, and physical interpretability.

The paper introduces RePPIC-Net, a hybrid AI framework that fuses 3D atmospheric physics from the FuXi model with geostationary satellite observations to quantify surface precipitation phase. This system represents a "first-ever" achievement in satellite-based nowcasting, offering a scalable template for global early warning systems.

The following key enhancements and justifications characterize this final version:

1. The authors have convincingly resolved concerns regarding operational latency. By initializing the FuXi model with ECMWF Operational Products, the system bridges data lags to generate 0-3 hour nowcasts with a total computational time of approximately 2 minutes.
2. The evaluation framework was significantly expanded to an 11-month period covering two complete winter seasons (2023-2025). Accuracy at the 2,000 national standard meteorological stations in China.

I recommend an acceptance of this work based on its impact during wintertime in data-sparse regions, and operational qualities.

Response: We thank you for your thoughtful assessment and for recognizing

the improvements made to the manuscript. We are pleased that the clarifications regarding operational latency, evaluation framework, and physical interpretability have strengthened the paper.

Reviewer 2 (Remarks on code availability):

The code mainly provides the ML architecture which may be useful for others to derive their model training methodologies from it.

Response: We agree that providing clear and accessible code is important for reproducibility and community use. We have **updated the README file** in the repository to better describe the ML architecture, training workflow, and how users can adapt the framework for their own data or regions. Also, the code repository has also been **archived on Zenodo** to ensure long-term availability and citability. The DOI is:

<https://doi.org/10.5281/zenodo.18357366>

We believe these steps enhance the utility and transparency of the code for researchers interested in applying or extending the RePPIC-Net framework. Thank you once again for your supportive and encouraging recommendation.

Reviewer Report

Summary:

The manuscript addresses the real-time monitoring of precipitation phase and intensity by integrating AI-driven atmospheric forecasts (FuXi), geostationary satellite observations (FY-4B), and auxiliary geolocation information. The authors employ a U-Net architecture for phase discrimination and a ResU-Net for intensity estimation, trained using GPM DPR products as reference labels. The topic is timely and relevant for advancing real-time precipitation monitoring, especially over data-sparse regions such as the Tibetan Plateau.

However, the manuscript has **major weaknesses in terms of novelty, methodological rigor, and the positioning for a high-impact science journal of *Nature Communications***. Below are the specific points of evaluation.

Strengths:

1. **Timely Topic:** Real-time phase-intensity discrimination is highly relevant for both weather and climate communities. Performance evaluation over challenging environments such as the Tibetan Plateau adds practical value and scientific interest.
2. **Integration of Multiple Data Sources:** Combining AI-driven atmospheric forecasts (FuXi), geostationary observations (FY-4B), and probabilistic classification methodology shows an attempt toward an integrated solution.
3. **Operational Relevance:** Emphasis on low-latency capability (leveraging FuXi forecasts and FY-4B observations), and short-term precipitation nowcasting (1–3 hours lead time) enhances the practical utility of the system.

Major Concerns:

1. Novelty and Conceptual Contribution

The core methodological contributions appear incremental rather than transformative. The use of Unet and ResUNet architectures for precipitation phase and intensity estimation is well-established in remote sensing and computer vision domains, and binary cross-entropy loss is a standard approach for classification problems. Furthermore, the phase classification algorithm relies heavily on the existing GPM-IMERG PLP framework, raising concerns about whether the approach represents a novel methodology or a repurposing of an established algorithm. The study risks being perceived as an application note rather than a fundamental advance.

2. Dependence on Latency and Real-Time Feasibility

The proposed system claims to enable real-time phase discrimination by combining FuXi forecasts (with ERA5 initialization) and geostationary satellite data. However, ERA5 initialization introduces an approximate 8-hour latency, which contradicts the claim of real-time operation. This undermines the practicality and uniqueness of the proposed framework compared to operational systems. Additionally, the reliance on FuXi's 12-hour forecasts and the inherent uncertainties in short-term atmospheric model outputs could introduce systematic biases, although there are optimization corrections, it has not fully addressed in the manuscript. The authors should rigorously quantify how

forecast errors propagate into phase discrimination and intensity estimation, particularly under extreme or rapidly evolving weather events.

3. Validation Strategy and Robustness of Results

The study uses GPM DPR as the reference for both precipitation phase and intensity, but this raises several concerns. First, GPM DPR's revisit time and sampling limitations make it an imperfect ground truth for evaluating full-disk geostationary-based predictions. Second, the performance evaluation primarily focuses on statistical accuracy over large areas and special regions (e.g., the Tibetan Plateau), but there is limited analysis of failure cases or uncertainty quantification in heterogeneous terrains and convective systems. Furthermore, the validation appears superficial and lacks comprehensive metrics, and temporal consistency checks. Without robust uncertainty analysis, the reliability of the integrated system for operational purposes remains unclear.

4. Positioning Against Existing Systems

The manuscript compares the proposed system to FY-4B and GPM IMERG-Late products, but these comparisons are primarily descriptive and do not convincingly demonstrate a clear performance advantage in terms of accuracy, latency, or operational applicability. The added value of integrating FuXi AI-driven forecasts with satellite observations needs to be explicitly quantified against state-of-the-art precipitation phase/intensity retrieval algorithms. In its current form, the manuscript does not sufficiently justify why the approach is superior to existing multi-source fusion techniques, limiting its perceived novelty and impact.

5. Scientific Insight vs. Algorithmic Engineering

While the technical implementation is sound, the study reads more like an engineering solution for operational nowcasting rather than a contribution that advances scientific understanding of precipitation processes. The manuscript lacks physical interpretability and mechanistic insight. Does FuXi better capture the 0°C isotherm or wet-bulb temperature structure? How does this improvement manifest across different synoptic regimes, e.g., warm vs. cold precipitation systems, freezing rain events, orographic snowfall? The authors could strengthen its scientific contribution by analyzing how integrated atmospheric forecasts and satellite signals improve phase discrimination under challenging physical conditions (e.g., mixed-phase precipitation, topographic influence on snow/rain transition), or by proposing a novel interpretable AI architecture informed by atmospheric physics. Currently, the discussion of underlying physical processes and their interaction with the AI-driven framework is limited.

Minor Comments:

1. The overall organization of the manuscript requires further improvement. The manuscript's structure is somewhat disorganized, and the presentation lacks clarity. Some content is repeated across sections, giving the text a redundant or "AI-generated" feel. In addition, some section of the

manuscript is described in a way that is not sufficiently detailed or precise, making it challenging for readers to fully understand the methodology, assumptions, and experimental procedures.

2. Using binary cross-entropy for precipitation intensity estimation is conceptually flawed, as precipitation intensity is inherently a continuous variable with a strongly right-skewed and zero-inflated distribution. Converting it into discrete classes for classification and applying binary or multi-class cross-entropy (BCE) results in a loss of important continuous information. This discretization introduces sensitivity to arbitrary thresholds and can lead to calibration biases, ultimately limiting the model's ability to accurately represent the true intensity distribution.